

# The impact of biomass burning and aqueous-phase processing on air quality: a multi-year source apportionment study in the Po Valley, Italy

Marco Paglione[1], Stefania Gilardoni[1], Matteo Rinaldi[1], Stefano Decesari[1], Nicola Zanca[1,*], Silvia
Sandrini[1], Lara Giulianelli[1], Dimitri Bacco[2], Silvia Ferrari[2], Vanes Poluzzi[2], Fabiana Scotto[2], Arianna
Trentini[2], Laurent Poulain[3], Hartmut Herrmann[3], Alfred Wiedensohler[3], Francesco Canonaco[4], André S.
H. Prévôt[4], Paola Massoli[5,6], Claudio Carbone[7], Maria Cristina Facchini[1], Sandro Fuzzi[1]

[1]Italian National Research Council - Institute of Atmospheric Sciences and Climate (CNR-ISAC), Bologna, 40129 Italy
[2] Regional Agency for prevention, environment and energy (ARPAE) of Emilia-Romagna, Bologna, Italy
[3] Leibniz-Institut für Troposphärenforschung (TROPOS), Leipzig, 04318, Germany
[4]Laboratory of Atmospheric Chemistry, Paul Scherrer Institute, Villigen PSI 5232, Switzerland
[5]Aerodyne Research, Inc. Billerica, MA, USA
[6]MultiSensor Scientific, Inc., Greentown Labs, Sommerville, MA, USA
[7]Proambiente S.c.r.l., CNR Research Area, Bologna, Italy
*now at Department of Chemistry and Institute for Atmospheric and Earth System Research (INAR), University of Helsinki,
FI-00014, Finland

*Correspondence to*: Marco Paglione (m.paglione@isac.cnr.it)

**Abstract.** The Po Valley (Italy) is a well-known air quality hotspot characterized by Particulate Matter (PM) levels well
above the limit set by the European Air Quality Directive and by the World Health Organization, especially during the colder
season. In the framework of the Emilia-Romagna regional project "SUPERSITO", the southern Po Valley submicron aerosol
chemical composition was characterized by means of High-Resolution Aerosol Mass Spectroscopy (HR-AMS) with the
specific aim of organic aerosol (OA) characterization and source apportionment. Eight intensive observation periods (IOPs)
were carried out over four years (from 2011 to 2014) at two different sites (Bologna, BO, urban background and San Pietro
Capofiume, SPC, rural background), to characterize the spatial variability and seasonality of the OA sources, with a special
focus on the cold season.

On the multi-year basis of the study, the AMS observations show that OA accounts for an average 45±8% (ranging 33-58%)
and 46±7% (ranging 36-50%) of the total non-refractory submicron particle mass (PM1-NR) at the urban and at the rural
site, respectively. Primary organic aerosol (POA) comprises biomass burning (23±13% of OA) and fossil fuel (12±7%)
contributions with a marked seasonality in concentration. As expected, the biomass burning contribution to POA is more
significant at the rural site (urban/rural concentrations ratio of 0.67), but it is also an important source of POA at the urban
site during the cold season, with contributions ranging from 14 to 38% of the total OA mass.

Secondary organic aerosol (SOA) contribute to OA mass to a much larger extent than POA at both sites throughout the year
(69±16% and 83±16% at urban and rural, respectively), with important implications for public health. Within the secondary
fraction of OA, the measurements highlight the importance of biomass burning ageing products during the cold season, even




at the urban background site. This biomass burning SOA fraction represents 14-44% of the total OA mass in the cold season, indicating that in this region a major contribution of combustion sources to PM mass is mediated by environmental conditions and atmospheric reactivity.

Among the environmental factors controlling the formation of SOA in the Po Valley, the availability of liquid water in the
aerosol was shown to play a key role in the cold season. We estimate that organic fraction originating from aqueous reactions of biomass burning products ("bb-aqSOA") represents 21% (14-28%) and 25% (14-35%) of the total OA mass and 44% (32-56%) and 61% (21-100%) of the SOA mass at the urban and rural sites, respectively.

## 1 Introduction

Ambient air pollution represents the highest environmental risks for human health, leading to about 3 million premature deaths every year (WHO, 2016) due to the exacerbation of respiratory and cardio-vascular diseases especially in young kids and elderly people. In Europe atmospheric pollution is responsible for more than 400.000 premature deaths a year (EEA, 2016), with the largest share due to fine particulate matter ($PM_{2.5}$ and $PM_1$) exposure. Organic aerosol (OA) accounts for 20 to 90% of fine particle mass worldwide (Zhang et al, 2007), and for up to 50% (20-90%) of fine particle mass in Europe
(Putaud et al., 2010). OA global budget and atmospheric processing are still characterized by large uncertainties (Hallquist et al., 2009). A better knowledge of OA is essential to support effective air quality control and remediation measures.

OA is directly emitted by various sources, including traffic, other combustion sources and biogenic emissions, and can also be produced via secondary formation pathways in the atmosphere (Hallquist et al., 2009). In particular, our understanding of the formation mechanisms and evolution processes of secondary OA (SOA) is still largely uncertain.

Direct quantification of SOA in the ambient aerosol is challenging, but many recent studies have proved that oxygenated OA (OOA) determined by multivariate statistical analysis (e.g., positive matrix factorization, PMF) of OA fragmentation mass spectra is a good proxy of SOA (Zhang et al., 2007; Ulbrich et al., 2009). Therefore, OOA is widely used to study the abundance and formation mechanisms of SOA. Although several types of these OOAs were isolated in ambient aerosol everywhere (often representing more than half of the total OA (Zhang et al., 2007; Ng et al., 2010; Crippa et al., 2014)), their
link to a specific source or mechanism remains largely undetermined. This is a consequence of their complexity in terms of chemical and physical properties, and the difficulty of reproducing the real conditions in which SOAs are formed/transformed. As a result, traditional models often show substantial discrepancies in simulating SOA mass concentrations (Kleinman et al., 2008; Matsui et al., 2009) and oxidation states (Chen et al., 2011), especially in wintertime.

The Po valley, located in northern Italy, is amongst the most polluted areas in Europe (EEA, 2016). It is surrounded by the Alps to the North and North-West and by the Apennines to the South. The occurrence of frequent and prolonged low-wind periods and atmospheric stability conditions favour the accumulation of particulate and gaseous pollutants locally emitted,



especially during the cold months. The distinctive features of the Po Valley make it an interesting "laboratory" to study the development of POA and SOA concentrations in the ambient atmosphere.

The SUPERSITO project (www.arpae.it/supersito) is a comprehensive study of atmospheric particulate matter pollution in the Emilia-Romagna Region, encompassing the southern part of the Po Valley from the Po river to the Apennines. Overall, the project deals with chemical, physical and toxicological parameters of the aerosol and integrates them with epidemiological and medical assessments through interpretative models. Results about aerosol chemical characterization using offline techniques were presented by Ricciardelli et al. (2017).

Here we describe the results of HR-AMS PM1 measurements carried out during eight intensive measurement campaigns with a focus on OA source apportionment. Previous projects have investigated the properties of fine aerosols at urban, rural and regional sites of the Po valley, including their chemical features (Carbone et al., 2014; Putaud et al., 2002, 2010; Saarikoski et al., 2012), and main sources (Belis et al., 2013; Gilardoni et al., 2011; Larsen et al., 2012; Perrone et al., 2012). Further studies based on aerosol mass spectrometer (AMS) measurements have been conducted in the same area during specific field experiments with the aim of characterizing specific phenomena and seasonal features (e.g. fog events, cooking aerosols, biomass burning emissions, etc.) (Gilardoni et al. 2014; Decesari et al., 2014; Paglione et al., 2014; Dall'Osto et al., 2015). Nevertheless, systematic AMS observations in the Valley are available from very few studies. Bressi et al. (2016) using a 1-year long dataset of measurements by an Aerosol Chemical Monitor (ACSM) described the chemical composition and the organic PM1 sources of the north-west edge of the Po Valley at the rural background site of Ispra, 60 km northwest of Milan.

In this study, we analyze a multi-year dataset of high resolution measurements carried out at two different sites (Bologna and San Pietro Capofiume) exploring for the first time the spatial-temporal variability of OA sources, chemical features and formation/transformation processes in the southern part of the Po Valley. A special focus is dedicated to the interpretation of the main sources and formation/transformation processes of the SOA in the region active during the cold period.

## 2 Material and methods

### 2.1 Measurement field campaigns

Eight intensive observation periods (IOPs) were carried out over four years (from November 2011 to June 2014) at two different sites of the southern part of the Po Valley (Bologna, BO, urban background and San Pietro Capofiume, SPC, rural background). Figure 1 reports a map of the measurement sites and a time-line of the field campaigns carried out during SUPERSITO project. Bologna is located at the foot of the Apennines and is an important population basin for the region (400,000 inhabitants), impacted by significant industrial and agricultural activities, and crossed by several major highways. The BO measurements site is located at the National Research Council (CNR) Research Area (N 44°31'29'', E 11°20'27''). The rural background station of San Pietro Capofiume (SPC) is located in a sparsely populated flat countryside (N



44°39'15'', E 11°37'29'') surrounded by kilometers of flat lands in the southeast part of Po Valley, 30 km north-east of Bologna and is representative of the regional background. This site is used for many atmospheric characterization studies and research projects (Saarikoski et al., 2012; Paglione et al., 2014; Decesari et al., 2014; Sandrini et al., 2016).

During the four-year project, the intensive campaigns were programmed to account for the marked seasonality in both

sources and weather conditions of this region. Nevertheless, most of the SUPERSITO campaigns took place in the cold season (3 campaigns in fall and 2 in winter, out of 8 in total) when the highest PM levels are found. Similar to other continental sites, during fall-winter the reduced height of the Planetary Boundary Layer (PBL) and calm wind conditions favor the accumulation of pollutants and are responsible for the rise of PM concentration (Perrone et al., 2012; Stanier et al., 2012; Bressi et al., 2013). Another feature of the cold months in this area is the high relative humidity, which leads to fogs

and hazes (i.e., conditions of high aerosol liquid water content, ALWC). The consequence of these meteorological conditions on PM concentrations is twofold: it promotes both wet removal (Gilardoni et al., 2014; Giulianelli et al., 2014; Montero-Martinez et al., 2014) and aqueous-phase processing with SOA formation (Gilardoni et al., 2016).

**2.2 Aerosol mass spectrometer measurements and apportionment of organic fraction**

During all of the SUPERSITO campaigns, the mass loading and the size-resolved chemical composition of submicron aerosol particles were obtained online by the Aerodyne high-resolution time-of-flight aerosol mass spectrometer (HR-TOF-AMS, Aerodyne Research, Canagaratna et al. 2007). The HR-TOF-AMS provides measurements of the non-refractory sulfate, nitrate, ammonium, chloride, and organic mass of the submicron particles (NR-PM1). The average concentrations of NR-PM1 chemical components and their relative contributions as measured by AMS in each campaign are reported in the

Supplemental material (Table S1 and Figure S1). For some of the SUPERSITO campaigns, specific studies have already been published. We refer to Gilardoni et al. (2014 and 2016) for the SPC fall 2011 and BO winter 2013 campaigns, respectively, and to Sullivan et al. (2016) for the SPC summer 2012 campaign. In this paper we focus on the organic aerosol (OA) component that represents the major fraction of submicron particles for most of the campaigns, ranging between 33 and 58% of NR-PM1 (concentration range: 1.8-18.4 µg m$^{-3}$), consistent with the value found by Jimenez et al. (2009), Ng et

al. (2010) and Crippa et al. (2014). Table 1 summarizes the average OA concentration for each site and season and the relative organic contribution to the NR-PM1 as measured by the HR-TOF-AMS.

The working principle of the HR-TOF-AMS is described in detail in Canagaratna et al. (2007), Jayne et al. (2000), and Jimenez et al. (2003). Briefly, during all the campaigns, the HR-TOF-AMS was operating by alternating between "V" and

"W" ion path modes every 5 min. The concentrations reported here correspond to the data collected in V mode. The resolving power (DeCarlo et al., 2006) of the V-ion mode was about 2000-2200 during all the campaigns.

Ionization efficiency (IE) calibrations were performed before and after every campaign, and approximately once a week during the campaigns. Filter blank acquisitions during the campaign were performed once a day to evaluate the background



and correct for the gas-phase contribution. All data were analyzed using the standard ToF-AMS analysis software SQUIRREL v1.51 and PIKA v1.10 (D. Sueper, available at: http://cires.colorado.edu/jimenez-group/ToFAMSResources/ToFSoftware/index.html) within Igor Pro 6.2.1 (WaveMetrics, Lake Oswego, OR). The HR-TOF-AMS collection efficiency (CE) was calculated based on aerosol composition, according to Middlebrook et al. (2012) and

evaluated against parallel offline measurements (see section 2.3 and Table S2 in the Supplement). The aerosol was dried to about 35-40% by means of a Nafion drier before sampling with the HR-TOF-AMS.

The organic fraction (OA) measured by HR-TOF-AMS was apportioned using the Positive Matrix Factorization approach (PMF; Paatero and Tapper, 1994; Lanz et al., 2007; Ulbrich et al., 2009; Zhang et al., 2011) by applying the Multilinear

Engine 2 solver (ME-2, Paatero, 2000) controlled within the Source Finder software (SoFi v4.8, Canonaco et al. 2013; Crippa et al., 2014).
Similarly to the classical PMF solver (e.g., PMF2, PMF3, Paatero and Tapper, 1994), the ME-2 solver (Paatero, 1999) executes the positive matrix factorization algorithm. However, the user has the advantage to support the analysis by introducing *a priori* information as known factor time series (FT) and/or factor profiles (FP), for example within the so-

called *a*-value approach. The *a*-value (ranging from 0 to values larger than 1) determines how much the resolved factors (($f_{k,j}$)solution, and ($g_{i,k}$)solution) are allowed to vary from the input ones ($f_j$, $g_i$), as defined in Eq. (1) (Canonaco et al., 2013).

$$(f_{k,j})\text{solution} = (f_{k,j})\text{reference} \pm a(f_{k,j})\text{reference}, \tag{1}$$

where $k$ and $j$ are the indexes for the factors and the species, respectively, $f_{k,j}$ is the element ($k,j$) of the F matrix (factor profiles, FP), the index "solution" stands for the PMF user solution, "reference" is a reference profile and "*a*" is a scalar defined between 0 and 1 (e.g applying an a value of 0.05 lets ±5% variability to our FP solution with respect to the reference FP during the PMF iteration). In our work we only constrained the mass spectra represented by f.

The standardized source apportionment strategy introduced in Crippa et al. (2014) is systematically applied to the 12 available HR-TOF-AMS datasets (8 from BO and 4 from SPC), consisting of the organic mass spectra over time and the corresponding errors.
The interpretation of the retrieved source apportionment factors as organic aerosol sources is based on the comparison of their mass spectral profiles with reference ones (Table S5, S6 and S7), on the correlations with external data (see Table S8)

and on the investigation of their diurnal trends. Details of the factor analysis (number of factors chosen, Q and residuals diagnostic plots, constrained factor profiles and *a*-values if applied) are reported for each campaign in the supplemental section S2.





### 2.3 Additional measurements and analytical techniques

Additional measurements from the routine daily program of the SUPERSITO project are used in this study as ancillary data. PM2.5 daily samples were collected by a low volume sampler (Skypost PM, TCR TECORA Instruments operated at the standard flow-rate of 38.3 L min-1) on quartz fiber filters (PALL Tissu Quartz 2500 QAO-UP 2500 filters, 47mm) during all

the project periods for the analysis of the carbonaceous fractions (total carbon, TC; organic carbon, OC; and elemental carbon, EC) by thermo-optical transmittance (Sunset, Laboratory Inc., Oregon, USA, using the EUSAAR2 thermal protocol, Cavalli et al., 2010; Ricciardelli et al., 2017) and of polar organic compounds (anhydrosugars and acids) by GC/MS analysis (Pietrogrande et al., 2014). Due to the elevated PM loading during the first experiment in fall 2011, the discrimination between OC and EC was not possible for the filters collected and only TC data are available for that specific campaign.

Black carbon (BC) was calculated from aerosol absorption coefficient measurements (when available) by a single-wavelength (573 nm) and a multi wavelength (467, 530, and 660 nm) Particle Soot Absorption Photometer PSAP (Bond et al., 1999), as previously described (Gilardoni et al., 2011; Gilarodni et al., 2016; Costabile et al., 2017).

Size-segregated aerosol particles were also sampled by a Berner impactor (flow rate 80 L min$^{-1}$ ) (Matta et al., 2003). The Berner impactor collects particles on five stages, corresponding to the following particle aerodynamic diameter cutoffs: 0.14, 0.42, 1.2, 3.5, and 10 μm. Sampling was performed continuously during the intensive campaigns. Each day we collected two samples: a daytime sample (from ≈ 09:00 to 17:00 LT during fall/winter, and from ≈ 9:00 to 21:00 LT during spring/summer), and a night-time one (from 17:00 to 09:00 LT during fall/winter, and from 21:00 to 09:00 during

spring/summer). Particles collected were extracted in water and analyzed by means of evolved gas analysis and ion chromatography for quantification of the water-soluble Total carbon (TC) and the inorganic species. Elemental and chromatographic analyses of the filter samples are used to validate the AMS data for the main aerosol components (Org, $NO_3$, $SO_4$, $NH_4$ and Cl) and PMF factors, as reported in the Supplemental (Table S2 and Table S8).

Submicron particles were also sampled on prewashed and prebaked quartz-fiber filters (PALL, 9 cm size) using HiVol samplers (a dichotomous sampler Universal Air Sampler, model 310, MSP Corporation at a constant nominal flow of 300 L min−1 or, alternatively, a TECORA eco-highvol equipped with Digitel PM1 sampling inlet, nominal flow 500 L min−1) located at ground level. Typically, two filters were collected every day in parallel with the Berner impactor sampling time. The HiVol quartz-fiber samples were analyzed to identify organic molecular tracers (e.g., levoglucosan,

hydroxymethansulfonate (HMSA) and low-molecular weight amines) using proton nuclear magnetic resonance ($^1$H-NMR) spectroscopy according to Decesari et al. (2006). The concentration of the organic tracers identified by NMR are correlated with the PMF-factors identified by the AMS, trying to detail their chemical features and infer their sources and atmospheric processing (especially for the OOAs).



Meteorological data are provided by the Hydro-Meteo-Climate Service of the Regional Environmental Protection Agency of Emilia Romagna (ARPAE). In addition, aerosol liquid water content that is associated with the aerosol inorganic species was predicted by the ISORROPIA-II model (Fountoukis and Nenes, 2007).

## 3 Results and discussion

### 3.1 Organic aerosol source apportionment

The source apportionment procedure allowed the identification of various components tracing the contributions of primary and secondary organic aerosol sources: hydrocarbon-like organic aerosol (HOA) resulting from the combustion of fossil fuels (e.g., vehicular traffic); BBOA (biomass burning organic aerosol) resulting from biomass combustion, mainly associated to wood combustion for domestic heating; COA (cooking organic aerosol) associated with specific food cooking practices. The latter is found just as a minor component of OM and only in one campaign at BO (spring 2014). The rest of the mass of sub-micrometer organic aerosol consists of oxygenated organic aerosols (OOAs), representative of secondary formation and/or ageing processes in the atmosphere. Factor analysis extracted different types of OOA with distinct time trends and/or spectral features. In this section, we will consider the OOA factors as a whole, while in Section 4 we will discuss a source attribution for the individual factors.

Figure 2 shows the average mass spectra of all the identified HOA (n=12), BBOA (n=10) and SOA (n=12) (reduced from high resolution, HR, to unit mass resolution, UMR, for better readability) together with their standard deviation. The comparison between our profiles from the Po Valley and reference profiles is reported in supplemental section 2.1 in term of theta-angle ($\theta$) between the spectra (Kostenidou et al., 2009). The theta-angle is a metric for the similarity between two spectra ($\theta$ <15° good; 15°< $\theta$ <30° partial; $\theta$ >30° bad similarity).

The HOA profile is characterized by peaks corresponding to aliphatic hydrocarbons including m/z 27, 41, 43, 55, 57, 69, 71, etc. (Canagaratna et al., 2004). The median HOA profile in our study shows a good overlap (mostly $\theta$ <15°) with almost all the reference spectra compared, as expected for this type of source which is quite reproducible in terms of AMS spectral characteristics (Crippa et al., 2014). Among the HOA profiles found for the individual campaigns, only one (SPC fall 2011) shows low correlations with the others from this study and with the references. Such discrepancy must be due to the peculiar conditions during the campaign, as the numerous fog events strongly impacted the OA time trends and, in turn, also the ability of PMF to resolve sources profiles. The aerosol observations during the SPC fall 2011 campaign have been already thoroughly described by Gilardoni et al. (2014) and will be summarized later in the discussion.

Unlike the HOA, the BBOA profiles are more variable, in agreement with earlier findings (Grieshop et al., 2009; Heringa et al., 2011) showing that the biomass burning aerosol mass spectrum is strongly affected by burning conditions and types of





wood/biomass. Nonetheless, the deconvolved BBOA profiles show good similarities with many reference spectra from previous studies with their characteristic peaks at m/z 29 ($CHO^+$), 60 ($C_2H_4O_2^+$) and 73 ($C_3H_5O_2^+$), which are associated with fragmentation of anhydrosugars such as levoglucosan (Alfarra et al., 2007; Aiken et al., 2009).

The COA factor was identified without any constrain only during the BO spring 2014 campaign. Its spectral profile exhibits

good similarities with the correspondent reference spectra (Mohr et al., 2012; Crippa et al., 2013a). The presence of this COA factor reduced sensibly the model residuals in the central part of the day and it is therefore considered in the final solution.

The more oxidized factors (OOA) differ from each other for the fractional abundance of m/z 43 and 44 and for the intensity of other fragments such as 29, 60 and 73. The spectral characteristics of the specific OOA factors are discussed in Section 4.

The correlation parameters between the time trends of AMS organic factors and of atmospheric tracer compounds are reported in Table S8 The time series of HOA correlates with that of elemental carbon (EC) or black carbon (BC) and with that of NOx. The correlation with NOx points to major sources of HOA from traffic. The trend of BBOA concentrations instead correlates with the trend for levoglucosan (measured by off-line techniques: GC/MS or $^1$H-NMR) and with the

organic fragments at m/z 60 and 73, which have been previously shown as good markers for biomass burning (Alfarra et al., 2007; De Carlo et al., 2008; Aiken et al., 2009). The concentration ratios between POA factors and tracer compounds (e.g., HOA/BC, HOA/NOx, BBOA/levoglucosan, etc.) are reported in Table S9 and compared with literature ranges. The overall good agreement between these source-specific ratios and the literature ranges confirms our apportionment of POA components. The time trends of the OOA concentrations are contrasted with those of secondary inorganic species (i.e. $NO_3^-$,

$SO_4^{2-}$ and $NH_4^+$) and with the organic fragments at m/z 43 (Org_43 = $C_2H_3O^+$) and 44 (Org_44 = $CO_2^+$) generally exhibiting good correlations.

The identified factors daily trends (HOA, BBOA, COA and SOA) are shown in Figure 3. Median diurnal patterns are reported together with the 10[th], 25[th], 75[th] and 90[th] percentiles for each factor, for the lumped datasets from all SUPERSITO campaigns and separately for Bologna (BO) and San Pietro Capofiume (SPC).

The daily trends of each organic component exhibit consistent characteristics during all the campaigns. HOA presents a diurnal cycle characterized by two maxima corresponding to the rush hours (impacted by the greatest vehicular traffic) between 8-9 and 18-20, in agreement with the attribution of this fraction to traffic sources. This is especially evident at the urban site of Bologna compared to the rural one in which the concentrations of HOA are lower and rush hour signatures are

weak, as expected for a rural background site. BBOA is characterized by a daily cycle with a midday minimum and a night-time maximum. This behavior reflects the combination of two factors: the influence of the mixing layer height - which favors pollutant accumulation near the ground at nighttime - and the daily pattern of the emissions from domestic heating, increasing in the evening/night hours. The concentrations of COA exhibits a characteristic daily trend with two maxima corresponding to the hours of main meals, one in the central hours of the day (12-14) and the other in the evening (20-21,



more pronounced due to the shallow boundary layer after the sunset). Finally, OOA exhibits an almost flat daily trend, reflecting its regional nature or the influence of multiple secondary formation processes. Therefore, the weak diurnal trends of OOA were not informative of potential sources of SOA in this region.

**3.2 POA and SOA contributions, seasonality and spatial variability**

Table 2 summarizes the site-specific and campaign-specific contributions of OA components determined by AMS factor analysis (see also Figure 4). A few clear seasonal patterns can be identified especially for the Bologna urban site for which a higher number of measurements are available (Figure 4).

In Bologna, HOA contributes for 11-18% of the mass of sub-micrometric OA in fall-winter and for 6-12% in spring-
summer. The slightly lower average HOA contribution during warmer season likely reflects the combination of two aspects: the reduction of work and school activities in summertime nearby the sampling areas, leading to a reduction of traffic emissions, and a possible meteorological effect due to the higher mixing-layer, resulting in an enhanced dilution of the primary pollutants locally emitted.

The contribution of BBOA varies instead from 17-38% in the fall-winter campaigns to 0-14% in summer-spring. In
particular, the contribution of BBOA has not been detected in the summer period analyzed by itself. Biomass burning therefore dominates over fossil fuel combustion as a source of primary organic aerosols at the urban site during the cold season. At the same site, OOA contributes for 44-68% of the mass of sub-micrometric OA in fall and winter, while its contribution in spring and summer period increases to 74-92%. The higher relative contribution of SOA in the warm period is expected given the reduction of residential combustion and the increased photochemistry. However, the OOA fraction in
the cold season is still quite high, considering the latitude and climate of Bologna, where sunshine duration in winter is less than 3 h per day  (in contrast to the almost 9 h in the summer). A discussion about SOA formation mechanisms alternative to gas-phase photochemistry is presented later in section 4.2.

At the rural site of San Pietro Capofiume, as expected, the dominant contribution to POA in the cold periods is provided by BBOA (varying between 28 and 33% of total OA mass during 2013 and 2011 fall campaigns, respectively) and the fraction
of OOA to total OA is larger than at the urban site (35-65% in fall, and reaching 96% in summer). Peculiar results were found for the SPC fall 2011 campaign, during which very large contributions of POA were recorded: the HOA fraction reached 32% of OA mass, somewhat strange for a rural site. This was likely due to the occurrence of persistent fogs scavenging the most water-soluble OA components and leaving the interstitial aerosol enriched in its most hydrophobic organic components (HOA) (Gilardoni et al., 2014).
A summary of the seasonality of OA fractions at the two Po Valley sites is shown in Figure 5. The COA fraction, that was determined only at BO during one individual campaign and in small amounts, was not considered here to simplify the comparison between the other components. The SPC fall 2011 campaign was also not included in this statistic since the



aerosol composition and concentrations for this experiment referred to a mixture of total OA and interstitial OA in fog conditions, as described above.

Table S10 reports the correlation coefficients between the PMF factors discussed so far and the main chemical species constituting the sub-micrometric aerosol masses measured by the HR-TOF-AMS. The highest correlations are observed

between OOA and secondary inorganic species, nitrate and ammonium sulfate, confirming the secondary nature of this fraction of OA. In particular, it can be noticed that OOA correlates better with ammonium nitrate in winter and fall, and with ammonium sulfate in summer and late spring, in agreement with previous results (Zhang et al., 2011).

For the campaigns carried out in parallel at the urban and rural site (summer 2012, spring 2013 and fall 2013), we estimated

an "urban increment", i.e., the increase in OA-type concentrations in urban areas with respect to the regional background. We expressed the increment as the ratio between the campaign average concentrations at the urban vs rural site, accordingly to season and the specific OA fraction considered (see Table 3). For total organic aerosol (OA) and for its OOA fraction, the ratios are quite constant throughout the seasons, varying between 1.13-1.36 and 0.97-1.30, respectively. By contrast, higher values were found for HOA (1.67, 1.91 and 2.85 in spring, fall and summer, respectively), in agreement with a major HOA

source from urban traffic. The urban increment of BBOA is less clear: it varies a lot between spring (in which its value is very high, i.e. 5.87) and fall (with 0.67). Nevertheless, the spring value is affected by the low and intermittent high BBOA levels likely indicating very local sources. The fall value seems more representative and suggests a higher contribution of BBOA in the rural areas, probably due to the more spread use of fire-places and wood-stoves for domestic heating and to additional possible sources such, as agricultural burning.

### 4 SOA sources and their evolution

In the previous section we presented OOA as one single component; however, the HR-TOF-AMS statistical analysis identified various OOA types that may indicate different formation (sources) and transformation processes (aging) of SOA in the aerosol. The number of OOA categories identified during the SUPERSITO campaigns ranged from one (for SPC fall

2011 campaign) to four (for SPC summer 2012 campaign). Most of the IOPs (7 out of 12) allowed the identification of three OOA factors.

The spectral profiles of the individual OOAs are distinguishable based on minor mass fragments and other parameters. Among the most common parameters used in literature for the distinction and interpretation of the various OOA factors, are: elemental ratios (OM:OC, O:C and H:C), the carbon oxidation state (OSc) and the fractional abundance $f$ ( where $f\#$ is the

ratio between the abundance of a specific ion and the total organic spectrum) of specific fragments in their spectral profiles (e.g., $CO_2^+$ at m/z 44 ($f44$); $C_2H_3O^+$ at m/z 43 ($f43$); $C_2H_4O_2^+$ at m/z 60 ($f60$); etc.). The elemental ratios and the relative proportion between $f43$ and $f44$ generally indicate the degree of oxidation and therefore the extent of aging of a single factor





(normally the less oxidized components exhibit higher H:C, lower O:C and less $f43$ and $f44$, while OOAs have O:C and $f44$ increasing with their degree of oxidation and aging in the atmosphere, largely due to the formation of carboxylic acids during this process) (Ng et al., 2010; Duplissy et al., 2011).

Tables S12 and S13 show a summary of the parameters for the analysis and interpretation of all the factors identified by the PMF statistical analysis (including the different OOAs listed in order of their O:C ratios) during the campaigns of the SUPERSITO project. We focus on two aspects: the influence of biomass burning emissions on OOA components and the importance of the aqueous-phase processing in their formation and evolution. A more comprehensive analysis of the OOAs features of particular IOPs is object of specific publications (Gilardoni et al., 2014; Sullivan et al., 2016; Gilardoni et al., 2016; Zanca et al., in preparation).

### 4.1 Biomass burning influence on SOA

The products of cellulose pyrolysis, such as levoglucosan and similar species (i.e., mannosan, galactosan, etc., collectively called hereinafter "anhydrosugars"), generate mass spectra with an enhanced signal at m/z 60 and 73 due to the ions $C_2H_4O_2^+$ and $C_3H_5O_2^+$, which are therefore considered good tracers of wood combustion (Schneider et al., 2006; Alfarra et al., 2007). So, the parameter $f60$ (the ratio of the integrated signal at m/z 60 to the total signal of OA mass spectrum) is used as a marker to evaluate the influence of biomass burning emissions to the OA components (Cubison et al., 2011).

Fresh biomass burning emissions (BBOA factors) exhibit the highest content of anhydrosugars ($f60$). During atmospheric aging, the relative intensity of anhydrosugars signal decreases because of degradation and oxidation reactions. At the same time, atmospheric aging leads to the oxidation of the molecules, which corresponds to the increase of oxygenated fragments in the mass spectrum, the most intense of which is at m/z 44 ($CO_2^+$, $f44$).

The contribution of $f60$ on the different OA components of each campaign is represented in Figure 6 by points in the $f44$ vs $f60$ space (Cubison et al., 2011) together with those of some references from previous studies (Aiken et al., 2009; Ng et al., 2011; Mohr et al., 2012; Saarikoski et al., 2012; Crippa et al., 2014; Florou et al., 2017). The background level indicating no influence of biomass burning is represented in Figure 6 (panels a -c) by a grey shaded area. As additional reference of OA not influenced by biomass combustion, we also report the measurements carried out during the summer 2012 parallel campaign at the high altitude background station of Mount Cimone (Rinaldi et al., 2015).

Figure 6 (panels a-c) shows that the spectral features of the OOA factors from several campaigns are those typical of aged OA (large $f44$) but also indicate the presence of anhydrosugars above the background level. This suggests a variable influence of biomass combustion on the OOA factors.

Such OOAs factors influenced by biomass burning (OOAx_BB) represent a substantial mass fraction of the total OA during the fall-winter period (17-61% at the Bologna site and 14-35% at SPC). In the spring season, the biomass burning impact on OOA composition is much less evident ($f60$ closer to the background levels) but still representing 37% of the total OA, more than twice the contribution of POA at BO during the spring 2013 campaign.





Additional tests and details on the determination of the biomass burning influence on OOA components are discussed in the supplemental section S2.2.3.

### 4.2 Biomass burning oxidation pathways

The vertical axis in Figure 6 is controlled by the oxidation of the bulk OA while the horizontal axis by the anhydrosugars loss. Thus, depending on the relative rates of these processes, the slopes of the virtual lines connecting the primary factors (BBOAs) and the corresponding aged PMF factors (OOAx_BB) are expected to be different. We do see indeed that slopes vary in different campaigns. We also see that two OOA_BB factors detected during BO fall 2011 and winter 2013 campaigns are connected to the primary BBOA with different slopes in the f60 vs f44 space (as reported by the arrows in

Figure 6a). This variability could suggest that the two OOA_BB components, observed during the same experiment, are formed through different oxidation rates and pathways due to the variable environmental conditions.

In order to test this hypothesis, the evolution of the BBOA into OOAs is further analyzed for BO fall 2011 and BO winter 2013 in Figure 6d using the O:C and the hydrogen-to-carbon (H:C) ratios of the BBOA and OOAx_BB factors in the Van Krevelen (VK) diagram. The VK diagram is typically used to investigate the OA evolution during field and laboratory

experiments (Heald et al., 2010; Ng et al., 2011). The plot allows to remove the effect of physical mixing between secondary and primary aerosols, providing a clearer interpretation of the results. Aerosol aging has the overall effect of increasing O:C ratios. In the VK plot the H:C vs. O:C slope of 0 is equivalent to the replacement of a hydrogen atom with an OH moiety, whereas a slope of −1 indicates the formation of carboxylic acid groups (Ng et al., 2011). O:C and H:C values are reported for BBOA (triangles), OOA_BB factors (squares and circles). The slope of the line that links BBOA to the squares (i.e.,

OOAx_BB) is close to zero while the line linking BBOA to the circles (i.e., OOAx_BB-aq) is between −0.5 and -1, suggesting possible different oxidation pathways. The negative slope indicates that OOAx_BB likely formed from BBOA through formation of carboxylic acid moieties, suggesting photochemical oxidation processes driven by OH radical (McNeill, 2015). Conversely, OOAx_BB-aq formation (slope 0) took place in aerosol water (i.e., wet aerosol) through dark chemistry consistent with the hydroxyl group formation (Lim et al., 2010; Gilardoni et al., 2016).

### 4.3 Aqueous-phase chemistry in SOA formation

Figure 7 shows the variations in contributions of the two BB-influenced OOA factors identified during the BO fall2011 campaign as a function of RH, together with some other meteorological and chemical parameters. The aerosol liquid water content (ALWC), as calculated by the ISORROPIA-II model, and the hydroxymethanesulfonate (HMSA) were used to trace

the effects of aqueous-phase SOA formation. HMSA is formed by the reaction of sulfite and bisulfite with dissolved formaldehyde in droplets and deliquesced aerosols and is oxidized by ozone at concentration as low as 10 ppb (Kok et al., 1986; Facchini et al., 1992; Whiteaker and Prather, 2003). HMSA was detected by the HR-TOF-AMS (following the



estimation method presented by Ge et al., 2012) during all the campaigns, and its presence was confirmed by off-line H-NMR analysis of filter samples. ALWC and HMSA exhibit a strong increase as a function of RH during the campaign indicating the possible influences of aqueous-phase processing at high RH levels (Figure 7, panel a). At the same time, temperature and solar radiation (Figure 7, panel b) decrease as function of RH suggesting a reduction of photochemical

activity.

These ambient conditions result into a large increase in the contribution of OOA2_BB-aq, whereas the OOA1_BB concentration remained relatively constant (Figure 7c). Dividing the individual OOA fractions for the total POA concentrations (considered as a surrogate of the planetary boundary layer, PBL) in Figure 7d, we observed similar variations of the BB-SOA components with RH > 60%, further supporting the above conclusion.

Extending this analysis to all the campaigns (see also Figure S5), we identified at least one OOA factor originating from biomass burning through aqueous-phase processing (OOAx_BB-aq) in 8 out of 12 datasets (all fall and winter campaigns plus spring 2013). The correlations of all the OOAs with the aerosol liquid water content (ALWC) and the hydroxymethanesulfonate (HMSA) are summarized in Table 4 (and reported also in Figure8 for the OOAx_BB-aq factors).

The spectral profiles of these OOA_BB-aq factors originated from aqueous-phase processing (reported in Figure 8) are characterized by higher signals at m/z 29 ($CHO^+$) and m/z 58 ($C_2H_2O_2^+$) in addition to the more common m/z 43 ($C_2H_3O^+$), m/z 44 ($CO_2^+$) and m/z 60 ($C_2H_4O_2^+$) that characterized also the other BB-influenced secondary components. The OOAx_BB-aq factors spectra have also good similarities (4< $\theta$ angle <29, see Table S15) between each other and with the OOA spectra recorded after fog dissipation at SPC during fall 2011 (Gilardoni et al., 2016).

The conclusion that these components are affected by aqueous-phase processing is further supported by the correlations between the OOAx_BB-aq factors and some specific fragment ions. As shown in Table S16 all the aqSOAs identified during SUPERSITO campaigns are well correlated with $C_2H_2O_2^+$, $C_2O_2^+$ and $CH_2O_2^+$, which are typical fragments of methylglyoxal and glyoxal, that are precursors of SOA via cloud processing (Carlton et al., 2007; Altieri et al., 2008). We further stress the link between biomass burning and these aqSOA by looking at the correlations of these components with specific fragment

ions of aqueous-phase products of phenol and guaiacol emitted during the biomass burning (namely PhOH-OH, $C_6H_6O_2^+$, m/z 110.037; PhOH-2OH, $C_6H_6O_3^+$ at m/z 126.032; GUA-OH, $C_7H_8O_3^+$ at m/z 140.047; GUA-2OH, $C_7H_8O_4^+$ at m/z 156.042), already identified in previous laboratory studies (Yu et al., 2014). Moreover, considering the elemental composition of the OOAx_BB-aq (Table S12-S13 and Figure 8) we notice that their O:C ratios, calculated following the Ambient Improved (AI) method (Canagaratna et al., 2015), are similar (on average 0.82±0.09) to the AI O:C ratios obtained

from laboratory oxidation of phenolic compounds (0.89±0.10, Sun et al., 2010; 1.03±0.17, Yu et al., 2014) and from the laboratory-generated SOA from the photoxidation of organic precursors in the aqueous phase (0.89±0.13, Lee et al., 2011; Lee et al., 2012).



In conclusion, BB-influenced SOA formed by aqueous-phase processing (bb-aqSOA) identified during the SUPERSITO campaigns represents a substantial mass fraction of the total OA during fall-winter months (14-28% at Bologna site and 14-35% at SPC). This component is often more than half of the total SOA influenced by BB-emissions, while the other half undergoes photochemical oxidation pathways leading to OOAx_BB. Overall, our results support the importance in the Po

Valley of SOA formation by aqueous-phase processing of wood combustion reported by Gilardoni et al. (2016), extending the ambient observations of these phenomena to a larger dataset (Figure 9).

## 5 Conclusions

The SUPERSITO project constitutes the first extensive (multi-sites and multi-years) time-resolved aerosol chemical
experiment in the Po Valley. Eight intensive observation periods (IOPs) were carried out over the four years of the project (from 2011 to 2014) at two different sites (Bologna, urban background and San Pietro Capofiume, rural background) using a High Resolution Aerosol Mass Spectrometer (HR-AMS). The source apportionment of the OA allowed improving our understanding of aerosol sources, their chemical features and spatial-temporal variability in the region, one of the most important pollution hot spots in Europe. Considering the special focus of the project on the cold season (3 campaigns in fall
and 2 in winter, out of 8 in total) it was especially possible to investigate the wintertime SOA formation pathways, which are the less characterized and, for this reason, one of the most important missing processes in atmospheric chemistry and air quality models (Tsimpidi et al., 2016).

The possibility to compare the organic factors identified by the HR-AMS with additional chemical tracers measured in parallel by other advanced spectroscopic techniques (i.e., NMR) and more traditional ones (e.g., IC, GC/MS, OC/EC, etc.)
provided new insights on the detailed chemical structure and especially on the formation and ageing mechanisms of SOA.

On the multi-years basis of the project, OA represent on average 45±8% (33-58%) and 46±7% (36-50%) of the total non-refractory submicron particles (PM1-NR) at the urban and rural site respectively, within the range reported in literature for other European sites (Crippa et al., 2014) and the Asian regions (Hu et al., 2017; Li et al., 2015; Wu et al., 2018 for China & East Asia; Chakraborty et al., 2018 for India), and slightly less than the values reported for Southeastern US (50-75%, Xu et
al., 2015; Budisulistiorini et al., 2016). Among this fraction, primary sources (POA) are dominated by biomass burning (23±13%), especially at the rural site (SPC) whereas the fossil fuel combustion (12±7%) is higher in the urban background site (Bologna) where it also presents a marked seasonality. However, the biomass burning contribution to POA remains the most important source of POA also at the urban site during the cold fall/winter seasons. The BBOA contribution ranging 17-38% at Bologna during the fall/winter seasons is not far from the values reported for other European cities (10-40% in Paris,
Crippa et al., 2013b; 5-27% from the EUCAARI multi-sites study, Crippa et al., 2014) and United States areas (e.g., 15-33% for Southeast US, Budisulistiorini et al., 2016) and slightly higher than that of other highly populated and polluted



cities/regions of Asia (11-14% at Beijing, China, Sun et al., 2018; 10-20% at Kunpur, India, Chakraborty et al., 2018) where, however, other combustion sources (i.e., coal) contribute to the POA fraction.

The contribution of oxidized organic aerosol (OOA, used as a proxy for SOA) were found to be much higher than the primary ones, regardless of site and season with a multi-year average of 66% (44-92%; st.dev.=16%) and 71% (35-96%;
st.dev.=27%) of the total OA mass, at the urban and rural site respectively. The SOA dominance is also observed during winter at the urban site, where the SOA represents on average 56% (50-61%; st.dev.=8%) of the total OA mass. Within this SOA, the measurements highlight the dominant presence of biomass burning secondary components, even in the urban background. The HR-AMS data indicate that the OA mass contributions of this SOA factor influenced by wood-combustion was of the order of 14-44% which translates into biomass burning emissions representing the 31-82% of the OM mass in the
Po Valley during cold months (fall and winter). Significant contribution of aged BB emissions on the OA mass loadings has been already suggested by previous studies regarding the Po Valley (Saarikoski et al., 2012) and different European (Paris, France, Crippa et al., 2013b; Eastern Mediterranean, Bougiatioti et al., 2014; Athens, Greece, Stavroulas et al., 2018), Asian (Beijing, China, Hu et al., 2017 & Sun et al., 2018; Kunpur, India, Chakraborty et al., 2018) and American sites (South-East US, Xu et al., 2015; Budisulistiorini et al., 2016). However, studies reporting the identification and quantification in ambient
of specific BB-influenced OOA factors are still very limited (Gilardoni et al., 2016; Xu et al., 2017).

Our study also identified and quantified a particularly relevant role of the aqueous-phase processing in the formation and transformation of primary biomass burning emissions. Aqueous SOA (aqSOA) factors identified as OOAx_BB-aq represent on average 21% (14-28%) and 25% (14-35%) of the total OA mass at the urban and rural sites, respectively, highlighting the importance of aqueous-phase processing for SOA formation and transformation. Considering the widespread wintertime
occurrence of fog, low-level clouds and wet aerosols in many other highly-populated sites enclosed in orographic basins (Benelux and Ruhr district, Paris and London basins, Cermak et al., 2009; Californian Central Valley, Baldocchi et al., 2014; Yangtze River corridor, Niu et al., 2010; and Indo-Gangetic plain, Saraf et al., 2011), this study strongly suggests that aqueous-processing can be a major driver for secondary aerosol formation in wintertime at all these sites, with extremely important consequences on air quality policy at the global level.

These results suggest the importance of a continuous monitoring system for better characterization of biomass burning-driven pollution in the Po Valley area, using complementary measurements both routinely and through intensive campaigns in order to explore the importance of biomass burning on air quality and climate.





**Author contributions.** M.P., S.G., S.D. S.F., and M.C.F. designed the research; V.P., S.G., S.D., and M.C.F. organized the field campaigns; M.P., S.G., M.R., L.P., P.M., and C.C. carried out the experiments and processed AMS data; M.P., S.G., L.P., and P.M. performed the AMS PMF; L.P., F.C., A.S.H.P, contributed to the PMF discussion and correction; M.P., S.G., N.Z., S.S., L.G., D.B., S.F., F.S., and A.T. collected and analyzed the aerosol samples. M.P., S.G., M.R., and S.D. wrote the
paper. H.H., A.W., and S.F. contributed the scientific discussion and paper correction.

**Acknowledgments.** This work is supported by Emilia-Romagna Region "Supersito" Project (DRG 428/10; DGR 1971/2013) and European projects PEGASOS (EU FP7-ENV-2010-265148) and BACCHUS (EU FP7-ENV-2013-603445). Authors acknowledge COST Action CA16109 COLOSSAL. Authors are grateful to all the ARPAE Emilia-Romagna co-workers for their support and collaboration in the realization of the field campaigns. ISAC-CNR is particularly grateful to
Leone Tarozzi, Francescopiero Calzolari and all the other colleagues who collaborated in the preparation of the campaigns, the functioning of the instruments and the aerosol sampling in the field.

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



**Table 1: Average organic aerosol (OA) concentrations and its relative contribution to the NR-PM1 mass measured by the HR-TOF-AMS within each campaign.**

|  | BO | | SPC | |
|---|---|---|---|---|
|  | OA ($\mu g\ m^{-3}$) | OA/NR-PM1 | OA ($\mu g\ m^{-3}$) | OA/NR-PM1 |
| 2011 Fall | 15.85 | 46% | 9.30 | 50% |
| 2012 Summer | 7.16 | 58% | 5.27 | 49%? |
| 2012 Fall | 4.61 | 46% |  |  |
| 2013 Winter | 8.37 | 42% |  |  |
| 2013 Spring | 2.04 | 44% | 1.74 | 36% |
| 2013 Fall | 3.81 | 33% | 3.37 | 40% |
| 2014 Winter | 3.60 | 39% |  |  |
| 2014 Spring | 3.31 | 54% |  |  |

5 **Table 2: Relative (%) and absolute mass contribution ($\mu g\ m^{-3}$) of main organic aerosol components HOA, BBOA, COA and OOA for all the considered campaigns. BO = Bologna, SPC = San Pietro Capofiume.**

|  |  |  | HOA | BBOA | COA | OOA |
|---|---|---|---|---|---|---|
| BO | SPRING | 2013 | 12% (0.25) | 14% (0.29) | - | 73% (1.49) |
|  |  | 2014 | 6% (0.18) | 2% (0.06) | 8% (0.28) | 84% (2.71) |
|  | SUMMER | 2012 | 8% (0.58) | - | - | 92% (6.58) |
|  | FALL | 2011 | 18% (2.80) | 38% (6.05) | - | 44% (7.00) |
|  |  | 2012 | 16% (0.74) | 30% (1.37) | - | 54% (2.50) |
|  |  | 2013 | 11% (0.43) | 17% (0.64) | - | 72% (2.74) |
|  | WINTER | 2013 | 11% (0.88) | 28% (2.35) | - | 61% (5.14) |
|  |  | 2014 | 12% (0.43) | 38% (1.37) | - | 50% (1.80) |
|  |  |  |  |  |  |  |
| SPC | SPRING | 2013 | 9% (0.15) | 3% (0.05) | - | 88% (1.54) |
|  | SUMMER | 2012 | 4% (0.20) | - | - | 96% (5.06) |
|  | FALL | 2011 | 32% (2.93) | 33% (3.07) | - | 35% (3.29) |
|  |  | 2013 | 7% (0.23) | 28% (0.95) | - | 65% (2.20) |





**Table 3:** Urban increment, calculated as the ratio between the campaign average concentration in urban and rural site, for each season and OA fraction considered.

| Urban Increment | | HOA | BBOA | OOA | OA TOT |
|---|---|---|---|---|---|
| SPRING | 2013 | 1.67 | 5.87 | 0.97 | 1.17 |
| SUMMER | 2012 | 2.85 | - | 1.30 | 1.36 |
| FALL | 2013 | 1.91 | 0.67 | 1.25 | 1.13 |

**Table 4:** Correlation (Pearson coefficients, R) between OOAs components identified by PMF/ME-2 and some variables linked to aqueous phase: Relative Humidity (RH) of the air; Aerosol Liquid Water Content (ALWC) calculated by the ISORROPIA II model; hydroxymethanesulfonate (HMSA) estimated by AMS and (when available) NMR measurements. The shaded cells highlight the highest correlations with a color scale ranging from less to more intense orange as the R value increases. The light-blue shaded cells highlight the identified aqSOA factor. Gray cells indicate missing data.

| Season | Campaign | BO | RH | ALWC | HMSA (AMS) | HMSA (NMR) | SPC | RH | ALWC | HMSA (AMS) | HMSA (NMR) |
|---|---|---|---|---|---|---|---|---|---|---|---|
| SPRING | 2013_spring (may) | OOA1_BB | 0.53 | 0.56 | 0.12 | - | OOA1 | 0.38 | 0.65 | 0.36 | |
| | | OOA2 | -0.15 | 0.53 | 0.40 | - | OOA2 | 0.19 | 0.76 | 0.55 | |
| | | OOA3_BB-aq | 0.33 | 0.75 | 0.62 | - | OOA3 | -0.09 | 0.53 | 0.58 | |
| | 2014_spring (may) | OOA1 | 0.25 | 0.11 | -0.03 | - | | | | | |
| | | OOA2 | -0.09 | -0.07 | 0.40 | - | | | | | |
| | | OOA3 | 0.13 | 0.23 | 0.56 | - | | | | | |
| SUMMER | 2012_summer (jun-jul.) | OOA1 | -0.26 | 0.12 | -0.28 | 0.32 | OOA1 | 0.08 | 0.20 | | 0.33 |
| | | OOA2 | 0.38 | 0.31 | -0.47 | 0.13 | OOA2 | 0.50 | 0.79 | | 0.20 |
| | | | | | | | OOA3 | 0.15 | 0.38 | | 0.51 |
| | | | | | | | OOA4 | -0.27 | -0.22 | | 0.42 |
| FALL | 2011_fall (nov.-dic.) | OOA1_BB | -0.12 | 0.63 | -0.06 | - | OOA_BB-aq | 0.00 | 0.88 | 0.77 | 0.66 |
| | | OOA2_BB-aq | 0.43 | 0.82 | 0.58 | - | | | | | |
| | 2012_fall (oct.-nov.) | OOA1 | 0.00 | -0.01 | -0.04 | - | | | | | |
| | | OOA2_BB-aq | 0.26 | 0.83 | 0.70 | - | | | | | |
| | | OOA3_BB | 0.11 | 0.20 | 0.55 | - | | | | | |
| | 2013_fall (oct.) | OOA1 | -0.29 | 0.16 | -0.22 | 0.15 | OOA1_BB | -0.24 | -0.29 | -0.06 | -0.17 |
| | | OOA2 | 0.28 | 0.65 | 0.76 | 0.53 | OOA2_BB-aq | 0.07 | 0.63 | 0.68 | 0.70 |
| | | OOA3_BB-aq | 0.34 | 0.82 | 0.81 | 0.86 | OOA3 | -0.04 | 0.28 | 0.45 | 0.25 |
| WINTER | 2013_winter (jan.-feb.) | OOA1_BB | 0.06 | 0.43 | 0.24 | 0.23 | | | | | |
| | | OOA2_BB-aq | 0.33 | 0.73 | 0.68 | 0.47 | | | | | |
| | | OOA3 | 0.29 | 0.32 | 0.19 | 0.16 | | | | | |
| | 2014_winter (jan.-feb.) | OOA1_BB | 0.16 | 0.38 | 0.43 | 0.57 | | | | | |
| | | OOA2_BB-aq | 0.31 | 0.74 | 0.72 | 0.71 | | | | | |
| | | OOA3 | -0.11 | 0.63 | 0.44 | 0.60 | | | | | |



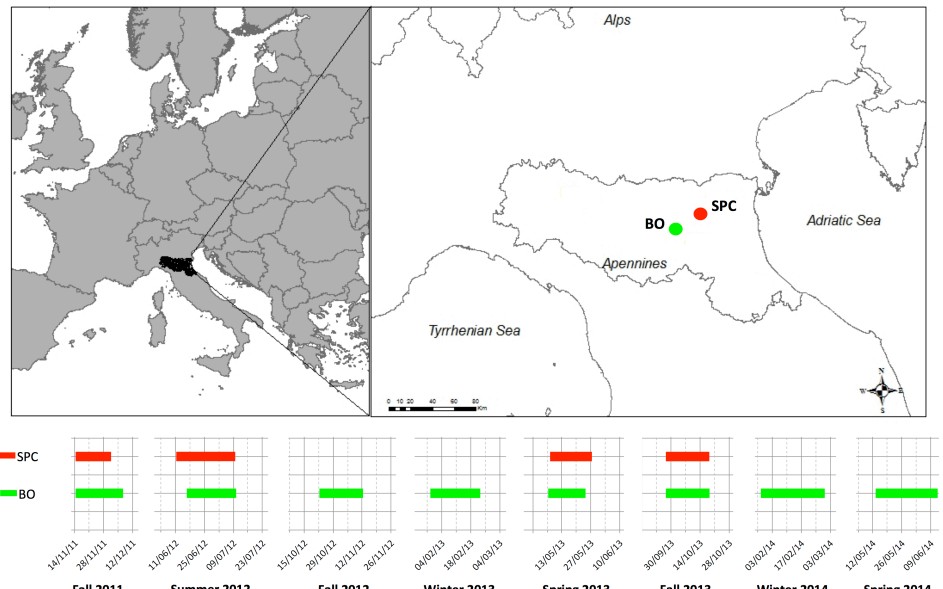

**Figure 1: SUPERSITO field campaigns: map of the sites and measurement periods considered in this study.**

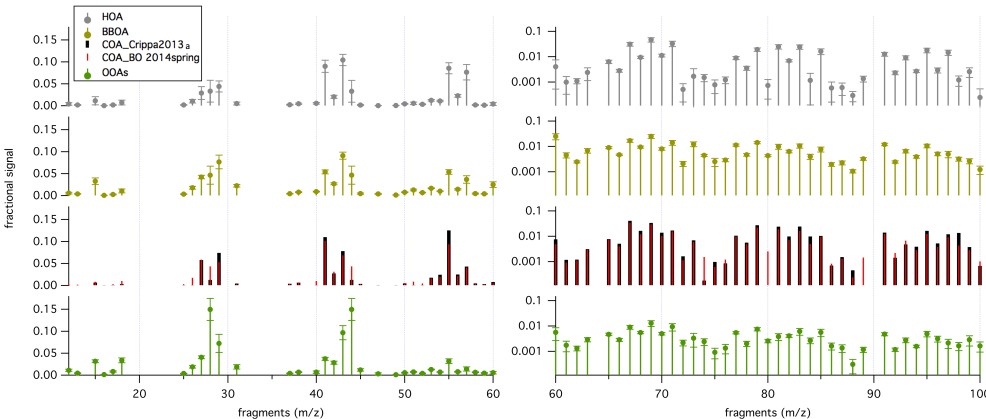

5 **Figure 2: Mass spectral variability for the main retrieved OA sources. Mean values are represented with circles and the ±standard deviation with error bars. COA from the BO spring 2014 campaign is represented in red color over-imposed to the COA reference spectrum from Crippa et al. 2013a.**





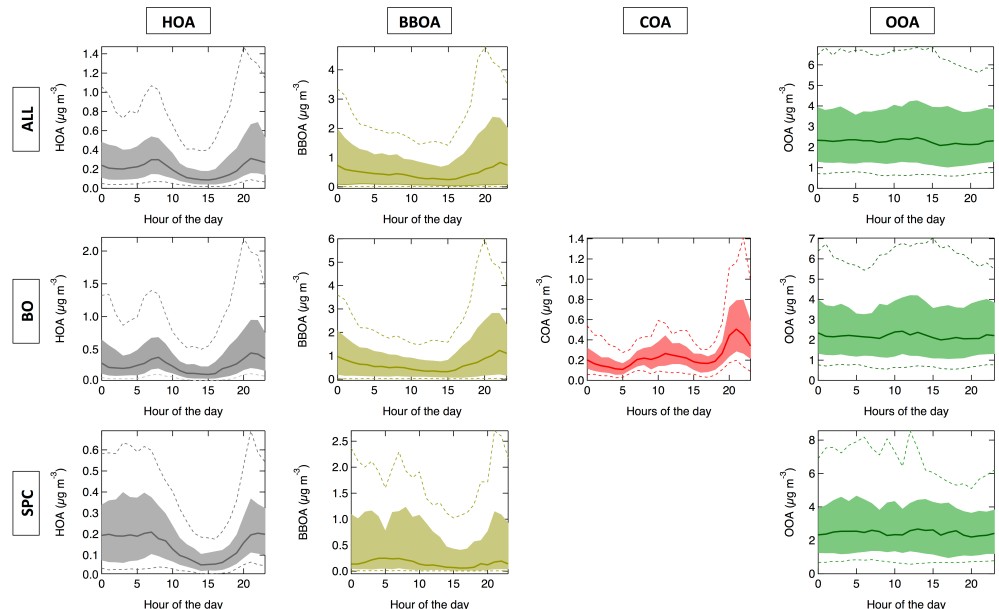

**Figure 3: Daily trends of the factors identified (HOA, BBOA, COA and SOA). Median diurnal pattern are reported together with the 10th, 25th, 75th and 90th percentiles for each source, for all the Supersito campaigns together and separately for Bologna (BO) and San Pietro Capofiume (SPC).**

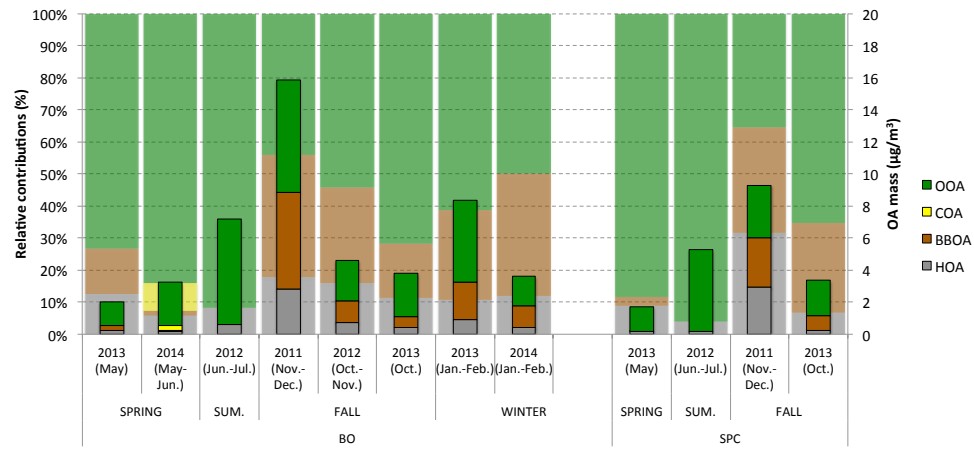

**Figure 4: Organic aerosol sources contribution for each site and each SUPERSITO campaign. Relative contributions are reported as shaded histograms (referring to the left axis) in the background of the absolute ones (referring to right axis).**




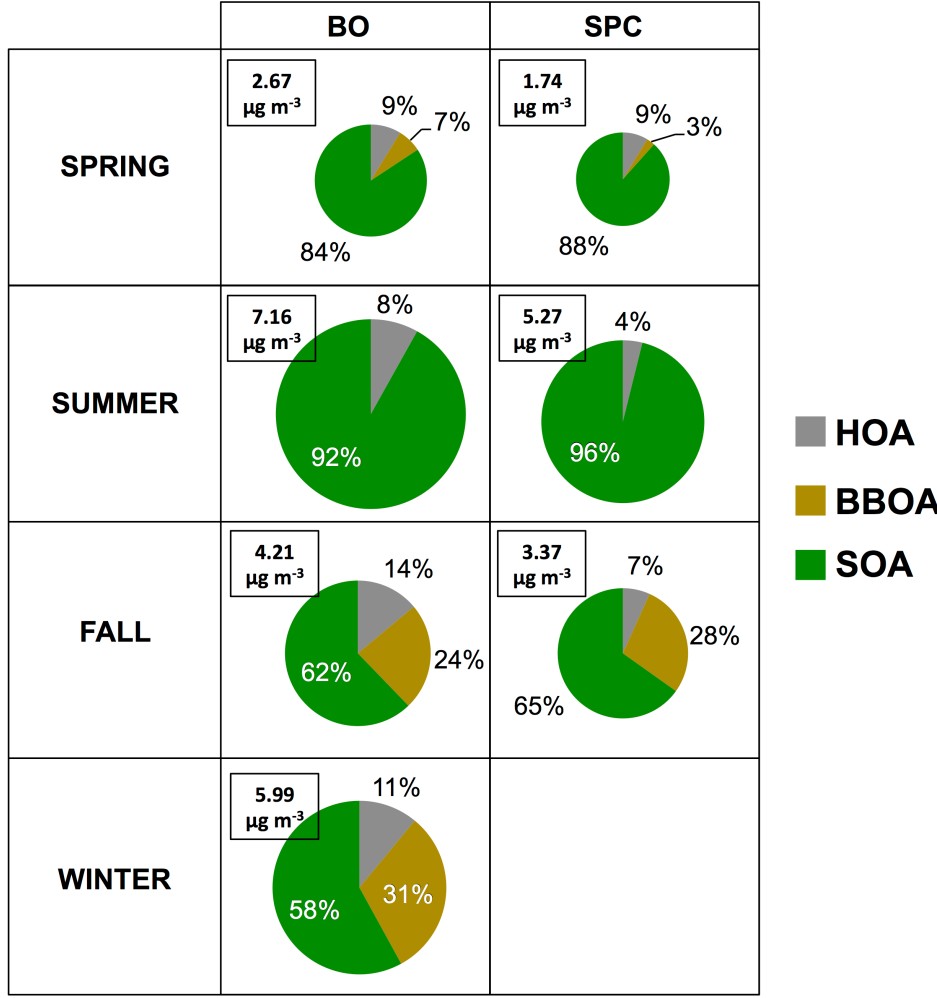

**Figure 5: Seasonal relative contribution of the main OA sources in both urban and rural site. Pie-charts area is proportional to the total average concentration of OA (reported in the upper-left side of each box in term of µg m$^{-3}$) and the individual portions are the average between the different campaigns made in the site in one season.**





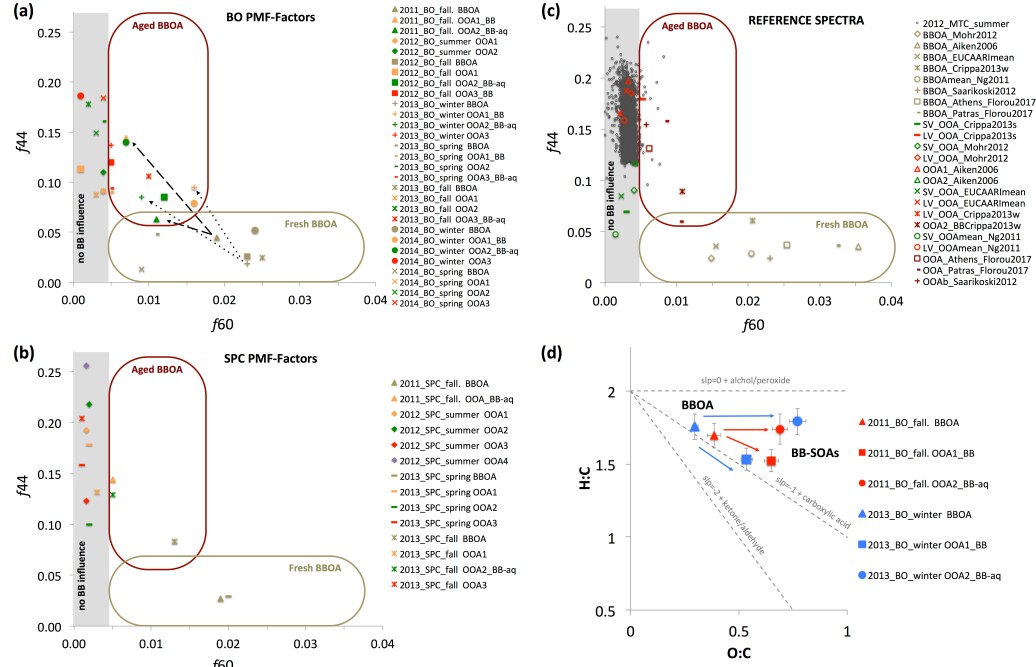

**Figure 6: Influence of Biomass Burning emissions on SOA and their evolution processes. The plots in panel a, b and c show $f44$ (normalized mass spectrum at m/z 44), which is a proxy of OA oxygenation degree, versus $f60$ (normalized mass spectrum signal at m/z 60), which is a proxy of anhydrosugars. Different shapes of the markers identify different SUPERSITO campaigns (panel a and b) or different reference spectra (panel c). Different colors represents the different kind of PMF-factors: gold-green identifies BBOA primary factors, yellow, green and red the OOAs numerically ordered based on their O:C ratios. Black dots in panel c) represent the measurements taken as background level of no influence of biomass burning. Gray areas correspond to $f60$ 0.003 ± 0.002 representing the Cubison et al. 2010 threshold of BB influence. Panel d) reports Van Krevelen (VK) diagram of the BBOA and OOA-BB PMF factors obtained from the HR-ToF-AMS data analysis for both BO fall 2011 (red markers) and winter 2013 (blue markers). The line connecting BBOA and OOA-BB has different slopes, indicating different chemistry processing leading to different SOA types.**





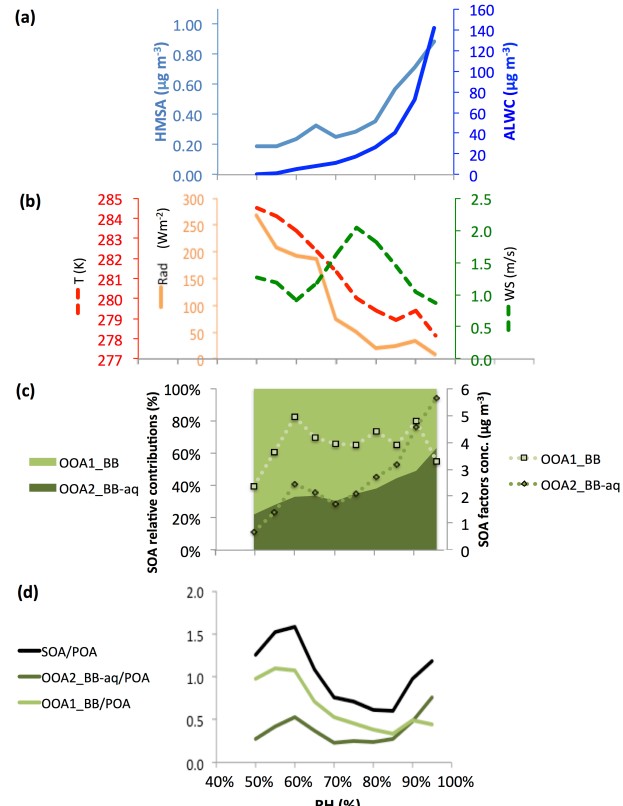

**Figure 7: variations of meteo and chemical parameters as function of RH during the BO fall 2011 campaign. The data were binned according to the RH (10% increment), and mean values are shown for each bin. Panel A: Aerosol Liquid Water Content (ALWC) and hydroximethansulfonic acid (HMSA). Panel B: air temperature together with solar radiation and wind speed (WS) measured at ground level. Panel C: variations in contributions of the two BB-influenced OOA factors identified (OOA1_BB and OOA2_BB-aq) both in absolute (µg m-3) and relative (% of OOA) terms. Panel D: different SOA categories excluding the effects of planetary boundary layer height (PBL) using the total POA as a surrogate.**



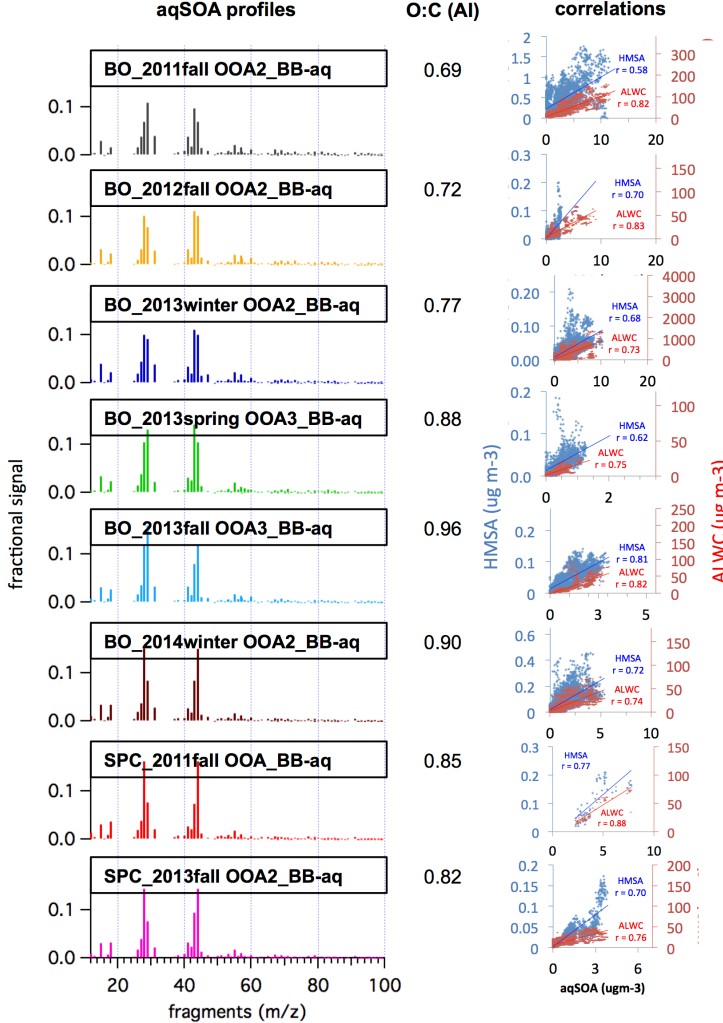

**Figure 8: OOAx_BB-aq main features: left column shows the mass spectral profile of each BB-aqSOA component identified during the SUPERSITO campaigns; central column reports the O:C elemental ratios of the same factors; right column illustrates the correlation between their concentration time series and the HMSA (in blue) and the ALWC (in red).**



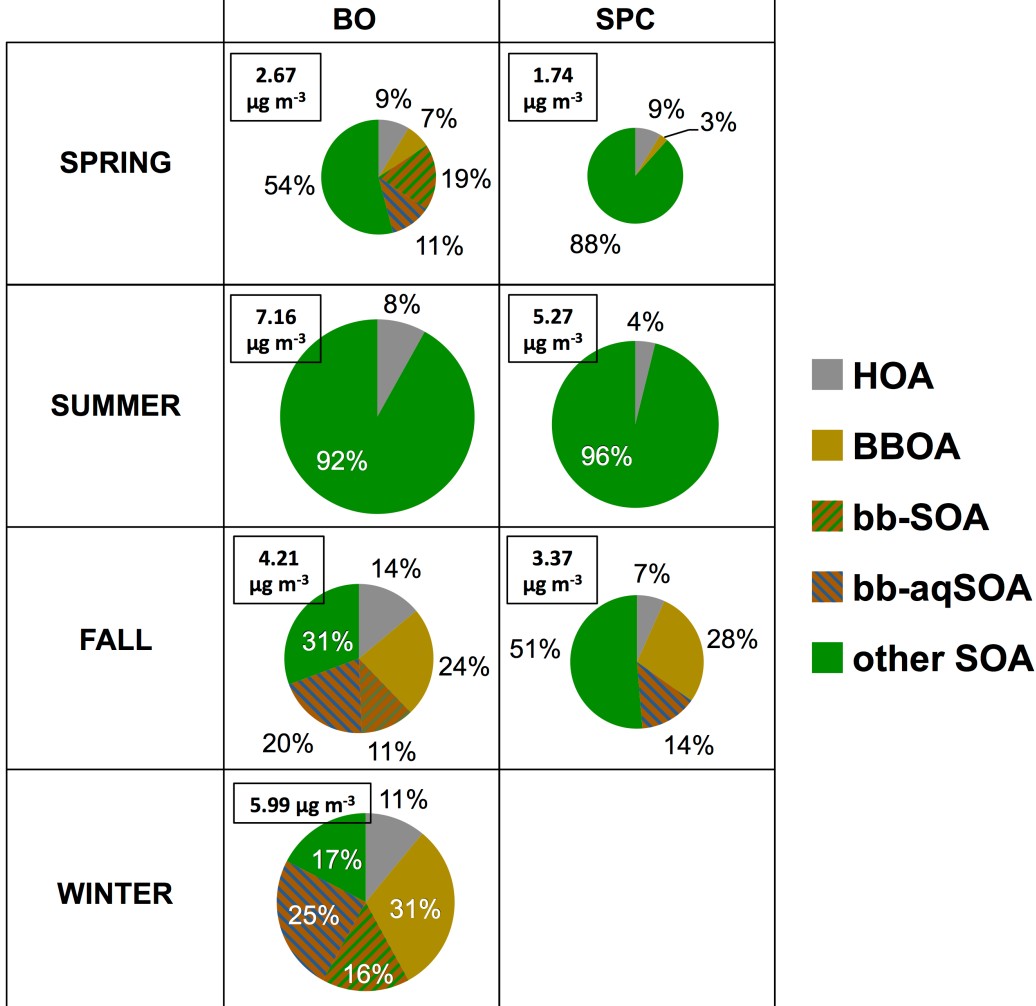

**Figure 9:** Seasonal relative contribution of the main OA sources in both urban and rural site with explicit separation also of the SOA (OOA) components. Pie-charts area is proportional to the total average concentration of OA (reported in the upper-left side of each box in term of µg m$^{-3}$) and the individual portions are the average between the different campaigns made in the site in one season. OOA factors influenced by biomass burning (characterized by brown background color) are divided in the two categories, "bb-SOA" and "bb-aqSOA" representing the OOAx_BB and OOAx_BB-aq described in the text . "Other SOA" is the sum of the other OOA factors whose source has not been unequivocally identified.