# Peer review of "The impact of biomass burning and aqueous-phase processing on air quality: a multi-year source apportionment study in the Po Valley, Italy"

_Atmospheric Chemistry and Physics, 2019_

## Referee Comment (RC1) · Anonymous Referee #1 · 3 Jun 2019

The paper by Paglione et al. presents a detailed analysis of AMS data obtained in several campaigns at Bologna and San Petro Capofiume throughout 2011 – 2014. The authors focus on source apportionment of OA by PMF and report a number of interesting findings, especially related to the impact of biomass burning and aqueous-phase processing. Generally, the methods and procedures applied are state-of-the-art, the paper is well structured and well written, and the results will certainly be welcomed with considerable interest by the readership of ACP. I have one major concern regarding the validity of "bb-aqSOA" identification (see below) and a few further minor issues. Overall, I recommend the paper for publication after the following issues have been addressed.

Main concerns: - Among the main findings of the study is the quantification of biomass burning SOA that is considered to result from aqueous phase reactions of biomass burning products (bb-aqSOA). This consideration, however, is mainly based on high correlations of one of the PMF OOA factors with ALWC. Even though plausible, I hesitate to take this correlation as sufficient evidence for really inferring the formation pathway of this OOA fraction, especially, as no alternative explanations are even discussed by the authors. One alternative explanation would be a strong shift in phase partitioning of water soluble organic gases during periods with high ALWC. A number of compounds like small carboxylic acids and (di)carbonyls present in the gas phase of (aged) BB plumes could be expected to preferentially partition into the particle phase only under high ALWC conditions, owing to their high volatility combined with high water solubility. These would contribute to the "OOAx_BB_aq" even without any aqueous phase chemistry taking place and OOAx_BB_aq might in fact just represent an OOA fraction with different phase partitioning behaviour than OOAx_BB", regardless of their respective formation pathways. There might be other possible explanations for the observed correlation with ALWC and I would recommend providing a more critical discussion on the subject.

- Further evidence provided to corroborate an aqueous phase formation pathway of OOA_BB_aq is either not conflicting with the above given alternative explanation or not fully convincing to me. Formaldehyde would preferentially partition into particles for high ALWC conditions to form HMSA (P13L2), taking AMS fragment ions as evidence for specific compounds in a complex ambient mixture might be questionable (P13L22 and P13L26) and many of these are well correlated to other OOAx factors as well (Table S16), the agreement with fog processing spectra is only partial in many cases (P13L18 and Table S15) and O:C ratios are unlikely to be good indicators for specific formation pathways (P13L29). To make myself clear: I am not in general disapproving the conclusion of an important aqueous phase bbSOA fraction, but I would encourage a more critical assessment of the evidence provided, including a discussion of alternative processes that could at least in part explain the observations.

Other issues:

- P5L18ff: In its present form, the explanation of the alpha value approach is not really accessible to non-familiar readers. Is it important to be presented or would a reference to the method suffice?

- P5L15 and P5L22: There is a contradiction in the range the alpha values can take.

- P5L23: "…represented by f." What does f mean?

- P6L21: "inorganic ions"?

- P6L23: ion charges are missing

- P9L15: "…in the summer period analyzed by itself" What does it mean? Please rephrase.

- P10L15: Are local sources important for the urban increment, i.e. does wind direction play a role for the high variability?

- P10L19: Is agricultural burning common in the area?

- P12L21ff: I do not follow these conclusions. Both carboxylate as well as hydroxyl group formation can in principle take place both in the gas and in the aqueous phase. The references cited all relate to aqueous phase chemistry. Please elaborate. Also, in Fig. 6d it is the circles that show a negative slope and in contrast to what the text says in L21 and L23, these are the ones labelled "BB-aq".

- P13L6ff: Given the very different emission strengths of POA during a day, I doubt it can be used as a surrogate of the PBL. Also, which "above conclusion" (L9) is supported by similar variations and why?

- P15L14: "ambient air"?

- P15L24: erase "extremely"

- Fig. 6 is hard to digest. Please structure the legend according to the different campaigns and increase panel sizes of a) – d). Maybe make d) a separate figure to gain space for a) – c)? Check readability of greyish color (really "gold-green"?) Consider using a consistent color code for OOAx_BB and OOAx_BB-aq. Right now, identifying the points in the scatter plot is a headache. Check the grey square for "no BB influence" for correct positioning. Text says 0.05 max, but the shaded area is < 0.05 in the plot.

- Fig. 6a) Some of the points are just at the edge of "BB influenced", based on f60. Is this considered in data interpretation and what does it imply for their assignment as "OOAx_BB"?

- Fig. 8: Check formatting of unit in axis labels.

---

## Referee Comment (RC2) · Anonymous Referee #2 · 18 Jun 2019

Review of the MS titled "The impact of biomass burning and aqueous-phase processing on air quality: a multi-year source apportionment study in the Po Valley, Italy" by Marco Paglione et al. MS id no: acp-2019-274 Air pollution in urban regions has gained considerable attention in recent years due to its health and climate-effects. Povalley (Italy) is one such hotspot region, where ambient PM levels are exceeding to that of both WHO and European Air quality Directive. Large uncertainties of this PM related health and climate-effects are somewhat associated with sources and processes effecting the organic fraction, which is a major part of ambient PM here. Paglione et al. have studied the intense air pollution events at two different sites (Bologna: BO and San Pietro Capofiume: PSC) over Povalley using High-Resolution Aerosol Mass

[Figure]

Spectroscopy (HR-AMS) with the specific aim of organic aerosol (OA) characterization and source apportionment over a period of four years (2011-2014). Overall, this is nice piece of study and worth publishing in ACP after addressing some of the below mentioned comments. I must say here that the uploaded text font of the ACP manuscript is too small read offline. I suggest authors to take care of this part when uploading the revision. Grey shades in figures should be in 'black' with increase in font size for all the figures. P2 L13: Is it 400 or 400,000 premature deaths?? P2 L20: replace 'proved' with 'established' P3 L16: Aerosol Chemical Monitor (ACSM)?? P9L25-29: I could not follow the logic of arguments here? Do authors mean HOA are embedded/occluded in water-soluble OA components at SPC, which are scavenged by the fog and left behind the HOA, thus increasing the fossil-based emission contribution to SPC?? Some additional explanation is needed here. From Table 1, it is apparent that OA contribute almost 50% at both sampling sites (BO and SPC). It is bit confusing to see some places OOA and other places as SOA. Please maintain consistency throughout the manuscript. In Table 2, I understand the reason of HOA share decrease between BO and SPC. But the BBOA component show more at BO site compared SPC during spring 2013 but also somewhat higher or comparable for other seasons too. Some explanation is need in the manuscript. P10 L8: Is it because of the differences in the ambient temperature and photochemical activity between winter and summer controls their abundance whether it is NH4NO3 in winter/fall vs. (NH4)2SO4 in summer and, hence, their correlation with OOA component. Add some additional explanations here. P10 L10: Instead of calculating based on the overage, I recommend authors' to show the ratio of each fraction of OA between BO and SPC based on the box plots. This will give us a brief idea about the relative increment of emissions/formation processes contributing to observed compound classes of OA between both sites. P10 L13: Authors mentioned previously that in summer traffic is less at BO because of the shutdown of schools and public institutions, in which case, why the HOA fraction increased over BO compared to SPC in summer. In Table 3, why BBOA fraction is almost 6 fold higher at BO (urban) compared to the SPC (rural). This implies there exists a very strong local

source of biomass combustion at BO compared to SPC, please clarify. Higher share of BBOA(%) over SPC in fall, why not is the case for winter or other seasons? P11 L6: Why focus only on these two factors? P11 L30-31: How this fraction of OOA_BB was estimated here? Figure 6 panel resolution and font sizes need to be improved? This figure is not readable at all offline. Grey shade text in panel d should be converted to black. P12 L19-20: These sentences are not clear, please rewrite. The slope line between triangles and circles seems to be zero (i.e., OOAx_BB-aq) and those between triangles and squares (OOAx_BB) is like between -0.5 and one. P12 L24: This is contradicting the above classification on L19-20. Please check. P12 L29: What are the input parameters to ISORROPIA-II, which mode is used, please provide. P13 L7: sentence should read like this, 'Dividing the individual OOA fractions with the total POA' In Figure 8, what is the OOA2, OOA3, OOA refers to, Please clarify. P13 L16: I can see m/z 29 signal but not 58 from figure 8 (left panel). Did I miss something here? P13 L21: mention those specific fragment ions here within parenthesis. In Figure 9, why there is no such presence of aq-SOA despite more sunlight and having precursors at both sites. Some explanations needed in the manuscript. Please provide line numbers continuously and also increase the font size of the text so that it will help us to properly evaluate the manuscript.

---

## Author Comment (AC1) · 9 Aug 2019

Replies to Referee #1

The authors would like to thank Anonymous Referee #1 for his/her comments.

The Referee's comments followed by our replies are listed below.

Main concerns: - Among the main findings of the study is the quantification of biomass burning SOA that is considered to result from aqueous phase reactions of biomass burning products (bb-aqSOA). This consideration, however, is mainly based on high correlations of one of the PMF OOA factors with ALWC. Even though plausible, I hesi-

[Figure]

tate to take this correlation as sufficient evidence for really inferring the formation pathway of this OOA fraction, especially, as no alternative explanations are even discussed by the authors. One alternative explanation would be a strong shift in phase partitioning of water soluble organic gases during periods with high ALWC. A number of compounds like small carboxylic acids and (di)carbonyls present in the gas phase of (aged) BB plumes could be expected to preferentially partition into the particle phase only under high ALWC conditions, owing to their high volatility combined with high water solubility. These would contribute to the "OOAx_BB_aq" even without any aqueous phase chemistry taking place and OOAx_BB_aq might in fact just represent an OOA fraction with different phase partitioning behaviour than OOAx_BB", regardless of their respective formation pathways. There might be other possible explanations for the observed correlation with ALWC and I would recommend providing a more critical discussion on the subject.

Authors Reply. The authors thank the Reviewer #1 for the critical view and suggestion. We acknowledge that a more clear explanation of our hypothesis can be profitable and it will be added in section 4.3 of the revised manuscript. We modified the manuscript in order to clarify that the attribution of OOAx_BB_aq factor to acqueous phase chemistry is not solely based on the correlation with ALWC. The consideration that the OOAx_BB-aq factors result dominantly from aqueous-phase reactions is still the most credible for two main reasons: 1- a technical one, the analyzed aerosol was dried to about 35-40% by means of a Nafion drier before sampling with the HR-TOF-AMS. So, most of the possible volatile compounds dissolved into particle phase under high ALWC conditions are expected to be volatilized again during sampling, leaving into aerosol only the most stable (complexed/oxidized by ageing) compounds; 2- linked with the previous, even if it's true that the partitioning of gaseous soluble species emitted by biomass burning is enhanced during high ALWC conditions, it is even true that these small and volatile compounds need to react to become stable components of particle phase. And this is exactly the mechanism that the Authors suggest is happening. The correlation of OOAx_BB-aq factors with HMSA is a good example of this mechanism,

as also the Reviewer #1 suggested in the subsequent comment: "Formaldehyde would preferentially partition into particles for high ALWC conditions to form HMSA" and ONLY if it happens the products of this reaction (i.e., HMSA) will remain in the particles after water evaporation. And this is supposed to happen to other gaseous species and their specific products, changing permanently the chemical composition of the organic aerosol.

- Further evidence provided to corroborate an aqueous phase formation pathway of OOA_BB_aq is either not conflicting with the above given alternative explanation or not fully convincing to me. Formaldehyde would preferentially partition into particles for high ALWC conditions to form HMSA (P13L2), taking AMS fragment ions as evidence for specific compounds in a complex ambient mixture might be questionable (P13L22 and P13L26) and many of these are well correlated to other OOAx factors as well (Table S16), the agreement with fog processing spectra is only partial in many cases (P13L18 and Table S15) and O:C ratios are unlikely to be good indicators for specific formation pathways (P13L29). To make myself clear: I am not in general disapproving the conclusion of an important aqueous phase bbSOA fraction, but I would encourage a more critical assessment of the evidence provided, including a discussion of alternative processes that could at least in part explain the observations.

Authors Reply. In addition to the previous reply, partially answering also to this comment, authors would like to add few more specific considerations: -HMSA is the product of sulfite complexation with formaldehyde, a reaction taking place only in aqueous-phase and inhibited by photochemistry. So the correlation of OOAx_BB-aq factors with this compound is in our opinion a clear evidence of the link with aqueous-phase chemistry and not only an indication of the preferential partitioning of formaldehyde at high ALWC conditions. - we acknowledge that the AMS fragments and elemental ratios are not overwhelming evidences. For this reason we first showed that the fragments that were attributed to HMSA correlates with the HMSA concentration, quantified also with independent measurements (H-NMR). In addition, in our study the other fragments are

used as ancillary evidences (for this reason reported in the Supplementary), not as the "main proof" that instead are the correlations with ALWC and HMSA and the variations with RH. -we acknowledge also that the variability of AMS spectra is high. However literature indicates that this can be quite common looking secondary factors (OOAs) but in any case they can be used to investigate SOA sources. This is clearly visible in Tables S7 and S14 for example where the comparison between OOAs factor spectral profiles from SUPERSITO campaigns and many correspondent reference profiles from literature is reported.

-P5L18ff: In its present form, the explanation of the alpha value approach is not really accessible to non-familiar readers. Is it important to be presented or would a reference to the method suffice?

Authors Reply. Authors thank the Referee and re-phrase the explanation of the a-value approach, removing the equation and summarizing the concept as follows: "Similarly to the classical PMF solver (e.g., PMF2, PMF3, Paatero and Tapper, 1994), the ME-2 solver (Paatero, 1999) executes the positive matrix factorization algorithm. However, the user has the advantage to support the analysis by introducing a priori information, such as known factor profiles (FP), for example within the so-called a-value approach. The a-value is a scalar (defined between 0 and 1) that determines how much the resolved factor profiles are allowed to vary from the reference ones (Canonaco et al., 2013). For instance, applying an a-value of 0.05 lets $\pm 5\%$ variability to our FP solution with respect to the reference FP during the PMF iteration."

-P5L15 and P5L22: There is a contradiction in the range the alpha values can take.

Authors Reply. Re-phrased and contradiction removed.

-P5L23: "…represented by f." What does f mean?

Authors Reply. Sentence removed rephrasing the concept.

- P6L21: "inorganic ions"?

Authors Reply. Replaced as suggested

- P6L23: ion charges are missing

Authors Reply. Ion charges added as suggested

- P9L15: "...in the summer period analyzed by itself" What does it mean? Please rephrase.

Authors Reply. Rephrased removing "analyzed by itself".

- P10L15: Are local sources important for the urban increment, i.e. does wind direction play a role for the high variability?

Authors Reply. In general, due to very low winds intensity (average values for all the campaigns of 1.89±0.32 and 2.06±0.17 m/s for BO and SPC respectively), the transport of pollutants along the two sites is supposed to be diffusive. So, the two sites are representative of two different background conditions (urban and rural), both poorly affected by really local sources in most of the cases (background sites) (http://www.arpae.it/sim/?osservazioni_e_dati/climatologia ; Riciardelli et al., 2017). The high variability affecting BBOA urban increment is something not expected, but (as expressed in the text, P10 L11, and showed in Section S2) probably it is due to the fact that the spring 2013 campaign was characterized by few nights with temperature colder (8°C) then the monthly average for May (18.5°C). These nights correspond to very sharp and intermittent peaks in BBOA concentrations (probably due to domestic heating active just for those few nights). For the rest, the BBOA concentrations during the campaign were really low (average values of 0.29 and 0.05 $\mu$m-3 at BO and SPC respectively), something not strange during May. So this very low and intermittent BBOA levels affect the urban increment value for this specific campaign and it is considered not representative.

- P10L19: Is agricultural burning common in the area?

Authors Reply. Actually agricultural burning is limited by the law: every Regional administration and sometimes also the different municipalities have a specific regulation on the topic especially during definite periods (e.g., dry summer, for fires prevention). However the open burning of agricultural waste is allowed for most of the year and continues to be a practice fairly common and difficult to regulate in rural areas (data confirmed by Regional Environmental Protection Agency of Emilia Romagna, ARPAE-ER). For this reason it is not possible to exclude the contribution of agricultural burning to the total BB aerosol burden at the rural background site (SPC), even if it is not possible to estimate its importance.

- P12L21ff: I do not follow these conclusions. Both carboxylate as well as hydroxyl group formation can in principle take place both in the gas and in the aqueous phase. The references cited all relate to aqueous phase chemistry. Please elaborate. Also, in Fig. 6d it is the circles that show a negative slope and in contrast to what the text says in L21 and L23, these are the ones labelled "BB-aq".

Authors Reply. Authors thank the Reviewer to highlight the discrepancies. We agree that in the present form the paragraph can be misleading for the reader. Our intention here is only to highlight the possible different oxidation pathways observed in the biomass burning ageing. Then in the subsequent paragraph we focus on the aqueous-phase chemistry. We rephrased the paragraph clarifying the message. We also add references of carboxylate formation not related to aqueous chemistry.

- P13L6ff: Given the very different emission strengths of POA during a day, I doubt it can be used as a surrogate of the PBL. Also, which "above conclusion" (L9) is supported by similar variations and why?

Authors Reply. Authors accept the Referee's suggestion: the sentence about POA as PBL surrogate is removed as well as the panel d) in Figures 7 and S5.

- P15L14: "ambient air"?

Authors Reply. Added

[Figure]

- P15L24: erase "extremely"

Authors Reply. Done

- Fig. 6 is hard to digest. Please structure the legend according to the different campaigns and increase panel sizes of a) – d). Maybe make d) a separate figure to gain space for a) – c)? Check readability of greyish color (really "gold-green"?) Consider using a consistent color code for OOAx_BB and OOAx_BB-aq. Right now, identifying the points in the scatter plot is a headache. Check the grey square for "no BB influence" for correct positioning. Text says 0.05 max, but the shaded area is < 0.05 in the plot.

Authors Reply. Done. A new version of the Fig. 6 is reported in the revised manuscript. The legend is now structured according to the different campaigns; the panel size is increased and the grey area for "no BB influence" is located in the right position. We prefer to leave the color distinction between different OOAs based on their O:C ratios and not on the formation process (which is instead highlight by the name of the factor). We prefer also to leave the panel d) in the same figure, even if increased in dimension.

- Fig. 6a) Some of the points are just at the edge of "BB influenced", based on f60. Is this considered in data interpretation and what does it imply for their assignment as "OOAx_BB"?

Authors Reply. As reported in the Supplementary section S2.2.3 (Validation of by Biomass Burning influenced OOAs) we performed additional tests in order to validate the attribution of the $C_2H_4O_2^+$ fragment (corresponding to the f60) to the OOA factors. In the end, factors considered as OOAx_BB are only those for which both average values and error bars (representing the standard deviation of all the additional tests performed) are located out of the gray shaded area indicating no influence of biomass burning (Fig. S3).

- Fig. 8: Check formatting of unit in axis labels.

Authors Reply. Done.

---

## Author Comment (AC2) · 9 Aug 2019

Replies to Referee #2

The authors would like to thank Anonymous Referee #2 for his/her comments.

The Referee's comments followed by our replies are listed below.

-I must say here that the uploaded text font of the ACP manuscript is too small read offline. I suggest authors to take care of this part when uploading the revision.

Authors Reply. ACP has specific rules about the submission format (e.g., font size, line numeration, etc.). The Authors simply followed these rules to satisfy the requirements

of the journal. The same is applicable for the last comment of the Referee #2: Please provide line numbers continuously and also increase the font size of the text so that it will help us to properly evaluate the manuscript.

-Grey shades in figures should be in 'black' with increase in font size for all the figures.

Authors Reply. We apply all the Referee's suggestions in the revised version of the paper.

-P2 L13: Is it 400 or 400,000 premature deaths??

Authors Reply. We thank the Referee to have notice the misspelled number. The correct number is four-hundred-thousand (400,000) of course. In order to avoid any confusion we completely remove the separator (putting 400000) in the revised version.

-P2 L20: replace 'proved' with 'established'

Authors Reply. Done.

-P3 L16: Aerosol Chemical Monitor (ACSM)??

Authors Reply. We thank the Referee to have notice the misspelling. The correct definition - Aerosol Chemical Speciation Monitor (ACSM) – is replaced in the revised version.

-P9L25-29: I could not follow the logic of arguments here? Do authors mean HOA are embedded/occluded in water-soluble OA components at SPC, which are scavenged by the fog and left behind the HOA, thus increasing the fossil-based emission contribution to SPC?? Some additional explanation is needed here.

Authors Reply. This behavior is better discussed in a previous publication by Gilardoni et al. (2014), specifically focused on SPC fall2011 campaign and cited at P9, L29. In the revised version we add a clearer linkage to this paper and we re-phrase the paragraph in order to clarify the concept, Here we just want to quickly explain the peculiarity of this specific campaign, representative of irregular meteorological conditions,

and so not completely comparable with the others in term of relative contributions of the different OA components. We would like to clarify that during the fall2011 campaign SPC was affected by persistent fogs during 14 days (out of the total 17). And (as reported in Gilardoni et al., 2014) fog scavenges aerosol components selectively, based on their solubility. For this reason, after fog formation, the interstitial aerosol (i.e., the fraction of the aerosol that is not scavenged and was measured in particle-phase) was dominated by particles enriched in carbonaceous aerosol, mainly black carbon and water-insoluble (or poorly soluble) organic aerosol. Moreover, analyzing the functional group composition and OA elemental ratios, Gilardoni et al. indicated that more oxidized OA was scavenged more efficiently than less oxidized OA. HOA is the less oxidized (O:C ratio = 0.29) and less soluble fraction of OA, so it is likely the one that is scavenged less efficiently and therefore its fractional contribution in the interstitial aerosol increases. So, the aerosol composition and concentrations for this campaign at SPC referred to a mixture of total OA and interstitial OA in fog conditions not fully comparable with the other campaigns.

-From Table 1, it is apparent that OA contribute almost 50% at both sampling sites (BO and SPC).

Authors Reply. Even if the comment is not clear, we think it refers to our sentence at P4, L22-23 "the organic aerosol (OA) component that represents the major fraction of submicron particles for most of the campaigns". We don't see any problem in this sentence: even when the OA fraction is less than 50% of the total mass there's not any other single component (NO3, SO4, NH4, etc.) accounting for a higher fraction of the PM1 mass. To clarify, we modified the sentence as follows: "the organic aerosol (OA) component that represents the most abundant fraction of non refractory submicron particles mass for most of the campaigns. . ."

-It is bit confusing to see some places OOA and other places as SOA. Please maintain consistency throughout the manuscript.

Authors Reply. We partially accept the suggestion of the Reviewer. In the revised manuscript we use always "OOA" when we are talking about the PMF factors. However sometimes we keep also the "SOA" abbreviation when we refer to more general arguments, like potential sources and formation pathways.

-In Table 2, I understand the reason of HOA share decrease between BO and SPC. But the BBOA component show more at BO site compared SPC during spring 2013 but also somewhat higher or comparable for other seasons too. Some explanation is need in the manuscript.

Authors Reply. Unfortunately we don't have enough reliable data to indicate a clear trend for BBOA concentrations between urban and rural site in all the seasons. The campaigns carried out in parallel at the two sites were only 4 (summer 2012, spring 2013 and fall 2011 and 2013). Among these: during the summer 2012 PMF didn't identify any BBOA (no domestic heating during summer); the fall 2011 had its meteo peculiarity linked with fog events (already discussed in a previous reply here above and in the revised text at P9, L21-26) and so it is considered not comparable; the spring 2013 was characterized by very low and intermittent BBOA levels (as better discussed in some other replies below) and so it is not clear how it is representative of background conditions. The only campaign reliable to say something about differences in BBOA concentrations between urban and rural site is the fall 2013, suggesting "a higher contribution of BBOA in the rural areas, probably due to the more spread use of fire-places and wood-stoves for domestic heating and to additional possible sources, such as agricultural burning", as clearly stated in the text.

-P10 L8: Is it because of the differences in the ambient temperature and photochemical activity between winter and summer controls their abundance whether it is NH4NO3 in winter/fall vs. (NH4)2SO4 in summer and, hence, their correlation with OOA component. Add some additional explanations here.

Authors Reply. We believe that there re two possible explanations. As suggested by

the Reviewer, the differences in temperature and relative humidity between winter and summer shift the partitioning of nitrate toward gas-phase (due to its volatility) during warm season. In addition, the different correlation suggests the possibility of a different oxidation pathways in secondary species formation between cold and warm season: a pathway characterized by cold temperature and high relative humidity (dominated by aqueous-phase processing and correlating more with nitrate) and another one more related with higher temperature and photochemical activity (correlating more with sulfate). We add in the revised manuscript a short explanation introducing the subsequent sections in which we developed more the concept.

-P10 L10: Instead of calculating based on the overage, I recommend authors' to show the ratio of each fraction of OA between BO and SPC based on the box plots. This will give us a brief idea about the relative increment of emissions/formation processes contributing to observed compound classes of OA between both sites.

Authors Reply. A more rigorous assessment of emission/formation processes would need a more detailed statistical treatment of the data variability between urban and rural sites, which is beyond the scope of the present manuscript. "Urban increment" here is not intended as a source apportionment tool but only as an instrument to discuss the differences observed between the urban and the rural sites. For this reason, even if we acknowledge the limitations of our urban increment assessment, we believe the calculation based on the average values is enough to have a rough idea of the most evident differences between the sites. In any case, we calculated the same ratios using Median value (reported in the table in Fig.1 here below), showing not substantial differences with Table 3. The high value corresponding to Spring 2013 (even higher than in Table 3) is consistent with the idea of few and intermittent high BBOA spikes, better discussed in the subsequent reply to the comment on Table 3.

-P10 L13: Authors mentioned previously that in summer traffic is less at BO because of the shutdown of schools and public institutions, in which case, why the HOA fraction increased over BO compared to SPC in summer.

Authors Reply. As the Urban increment clearly show (Table 3 and also previous comment), the HOA trend is constantly shifted toward higher concentrations at the urban site (BO) with respect to the rural one (SPC). This is quite expectable and already mentioned in the text: the urban site is closer to the HOA primary sources (traffic) with respect to the rural site and for this reason HOA concentrations are always higher at BO than at SPC, even when (during summer) are depleted (compared to the winter) due to the activities shutdown and increased dilution.

-In Table 3, why BBOA fraction is almost 6 fold higher at BO (urban) compared to the SPC (rural). This implies there exists a very strong local source of biomass combustion at BO compared to SPC, please clarify. Higher share of BBOA(%) over SPC in fall, why not is the case for winter or other seasons?

Authors Reply. Overall both the measurement sites are representative of background conditions: BO is representative of urban background, while SPC of the rural background. This because of two main reasons: 1- both the field stations are located quite far form any kind of direct local source and 2- due to very low winds intensity (average values for all the campaigns of 1.89±0.32 and 2.06±0.17 m/s for BO and SPC respectively), the transport of pollutants is supposed to be diffusive in the studied area (http://www.arpae.it/sim/?osservazioni_e_dati/climatologia ; Riciardelli et al., 2017). The high BBOA urban increment during spring 2013 is something not expected, but (as expressed in the text, P10 L11, and showed in Section S2) probably it is due to the fact that the spring 2013 campaign was characterized by few nights with colder temperature (8°C) then the monthly average for May (18.5°C). These nights correspond to very sharp and intermittent peaks in BBOA concentrations (probably due to domestic heating active just for those few nights). For the rest, the BBOA concentrations during the campaign were really low especially in SPC (average values of 0.05 $\mu$m-3), something not strange during May. So this very low and intermittent BBOA levels affect the concentrations for this specific campaign, which is considered for this reason not representative. About winter seasons: unfortunately our dataset is not comprising a winter

campaign carried out in parallel at both the sites and for this reason we can not argue nothing about the urban increment during winter.

-P11 L6: Why focus only on these two factors?

Authors Reply. Considering the big amount of data already reported in the paper, the Authors decided to focus more only on the two most innovative aspects emerged from this study, namely the biomass burning influence on SOA components and the aqueous-phase processing affecting some of these components. These two aspects are represented by the OOA_BB factors deeper examined, indeed.

-P11 L30-31: How this fraction of OOA_BB was estimated here?

Authors Reply. This fraction is represented by the mass contribution of the OOA factors which, looking the f60 vs f44 space, resulted influenced by biomass burning. As reported in the Supplementary section S2.2.3 (Validation of by Biomass Burning influenced OOAs) we performed additional tests in order to validate the attribution of the $C_2H_4O_2^+$ fragment (corresponding to the f60) to the OOA factors. In the end, factors considered as OOAx_BB are only those for which both average values and error bars (representing the standard deviation of all the additional tests performed) are located out of the gray shaded area indicating no influence of biomass burning (Fig. S3).

-Figure 6 panel resolution and font sizes need to be improved? This figure is not readable at all offline. Grey shade text in panel d should be converted to black.

Authors Reply. A new version of the Fig. 6 is reported in the revised manuscript.

-P12 L19-20: These sentences are not clear, please rewrite. The slope line between triangles and circles seems to be zero (i.e., OOAx_BB-aq) and those between triangles and squares (OOAx_BB) is like between -0.5 and one. P12 L24: This is contradicting the above classification on L19-20. Please check.

Authors Reply. Authors thank the Reviewer to highlight the discrepancies. The text was mixed up during writing and so the description of the figure is not consistent. We

rephrased the paragraph clarifying the message.

-P12 L29: What are the input parameters to ISORROPIA-II, which mode is used, please provide.

Authors Reply. Info provided in section 2.3 of the revised manuscript.

-P13 L7: sentence should read like this, 'Dividing the individual OOA fractions with the total POA'

Authors Reply. Authors, accepting a comment from Referee #1, removed the sentence about POA as PBL surrogate and removed also the panel d) in Figures 7 and S5.

-In Figure 8, what is the OOA2, OOA3, OOA refers to, Please clarify.

Authors Reply. OOA factors are numerically ordered based on their O:C ratios, independently on the influence of a specific source, as axplained in the text (P11,L5) and in the captions of Figure 6, showing for the first time in the paper the different OOAs. To improve clarity we add this information in the revised manuscript also in the caption of Figure 8.

-P13 L16: I can see m/z 29 signal but not 58 from figure 8 (left panel). Did I miss something here?

Authors Reply. The ion at m/z 58.01 01 ($C_2H_2O_2+$) is associated in literature to aqueous-phase reactions because is one of the typical fragments of precursors of SOA via cloud processing, like methylglyoxal and glyoxal (Carlton et al., 2007; Altieri et al., 2008). We state this clearly in the text (P13, L27-29 of revised manuscript). Since m/z 58.01 has much lower fractional abundances (f58) with respect to other fragments in the spectra this ion is not easily distinguishable in Figure 8 (where the spectra are reported in a very synthetic way). We acknowledge this but we prefer to avoid adding more information in Figure 8 (already quite packed). Anyway the f58 is always higher in all the OOA_BB-aq with respect to the others OOA_BB factors (as showed in the additional graph here below, Fig. AR1). This feature, associated with the abundance of

m/z 29 (CHO+), can help in the identification of the BB-aqSOA factors, even if it is not a sufficient proof (because the correlation with ALWC and HMSA together with the study of the variations with RH are more important to unambiguously identify those factors).

-P13 L21: mention those specific fragment ions here within parenthesis.

Authors Reply. They are mentioned in the subsequent sentences. The Authors don't see the need to add them also before within parenthesis.

-In Figure 9, why there is no such presence of aq-SOA despite more sunlight and having precursors at both sites. Some explanations needed in the manuscript.

Authors Reply. The question is hard to understand because no specific reference to the text or the figure is provided. Anyway, we think the comment is referring to the absence of a BB-aqSOA at SPC during Spring 2013. We believe it is probably related to the very low and intermittent BBOA levels affecting this specific campaign (already discussed above) and not allowing an important BB-SOA formation.
* * *
| Urban Increment using Median values | HOA | BBOA | OOA | OA TOT |
|---|---|---|---|---|
| SPRING 2013 | 1.73 | 8.85 | 1.28 | 1.42 |
| SUMMER 2012 | 2.94 | | 1.60 | 1.64 |
| FALL 2013 | 1.70 | 0.64 | 1.20 | 1.11 |

**Fig. 1.** Urban increment, calculated as the ratio between the campaign Median concentration in urban and rural site, for each season and OA fraction considered.

[Figure]

**Fig. 2.** Fractional abundance of m/z 58.01 (f58) in the spectral profiles of each OOA factor identified by PMF analysis in all the campaigns where a BB-aqSOA factor is identified. Blue bars represe

---

## Referee Report (RR1)

**Review: The impact of biomass burning and aqueous-phase processing on air quality: a multi-year source apportionment study in the Po Valley, Italy**

Paglione et al.

**General comments**

The manuscript by Paglione et al. focuses on a positive matrix factorization (PMF) on organic aerosol (OA) mass spectra recorded over four years with an high reslution time-of-flight aerosol mass spectrometer (HR-AMS) at two different sites in Po Valley, Italy. The study aims to quantify the biomass burning organic aerosol via primary and secondary formation mechanisms in the polluted Po Valley region, known to be greatly impacted by high pollution levels. The largest new discovery in the paper was the quantification of the biomass burning organic aerosol (BBOA) formation and processing in the aqueous phase that is clearly motivated by a previous publication by Gilardoni et al. published in PNAS in 2016. As a result of the PMF analysis by Paglione et al., a factor with distinct BBOA mass spectral fingerprints together with a high degree of oxygenation that correlates with aerosol liquid water content was linked to aqueous BBOA formation. The contribution of this factor to the total OA recorded ranged all the way up to 100% during the cold seasons, which makes these results of high importance and atmospherically relevant. While some parts of the study are fairly nicely written and formulated, I suggest few major additions and changes to be done before a review round for detailed adjustments should take place.

**PMF-analysis**

First, as PMF plays a central role in the aqueous BBOA quantification, the description of the PMF analysis should not by any means be hidden in the supplementary material and it should be thoroughly step by step explained. The PMF analysis process arises few concerns regarding a-values and the amount of repetitions performed (number of *iterations*) that seems not to be reported at all in the manuscript. I suggest you explicitly write down in the manuscript how many repetitions were conducted to make the readers aware of the statistical robustness of the solution. Currently, the vague description of the PMF analysis arises doubts of the solidness of the result.

For example, the justification of different a-values used in the multi-linear engine (ME-2) analysis is not clear. The high a-value used for HOA (a = 0.5) is exceptional, and I personally did not see why it was necessary. Previous studies suggest an a-value of approx. 10% for HOA. In contrary to this, the BBOA was constrained with an a-value of 5%. This is allows a rather low degree of variability for the BBOA mass spectrum in the PMF analysis despite the fact that BBOA is known to vary a lot depending on the burning material (even up to 30%). While the a-values chosen in the current study can in some cases be fully justified, the motivation still needs to be explicitly written down.

It is absolutely crucial that you carefully motivate the selections of the constraints (a-values) and the solutions (number of factors) in the manuscript main text. Please spend time on this to make sure your readers (and I) can follow your process. You can even create a flow chart to describe it, but try to avoid just transferring the table jungle from the SI to the main text, try to summarize the logic you had in decision making.

Here are two examples of publications where the PMF process is well documented:

Elser, Miriam, et al. "New insights into $PM_{2.5}$ chemical composition and sources in two major cities in China during extreme haze events using aerosol mass spectrometry." Atmospheric Chemistry and Physics 16.5 (2016): 3207-3225.

Daellenbach, Kaspar R., et al. "Long-term chemical analysis and organic aerosol source apportionment at nine sites in central Europe: source identification and uncertainty assessment." Atmospheric Chemistry and Physics 17.21 (2017): 13265-13282.

**$CO_2^+$ release from the AMS vaporizer due to ammonium nitrate**

Another important concern of mine is the $CO_2^+$ release from the AMS vaporizer coinciding with high ammonium nitrate loadings. This effect was introduced recently by Pieber et al. (2016) and investigated thoroughly with AMS-type instrumentation by Freney et al. (2019). My wonder is whether you quantified this effect with your instrumentation. How much $CO_2^+$ did you detect during your ionization efficiency calibrations? Did you modify your fragmentation table(s) and adjust the $CO_2^+$ introduced by ammonium nitrate?

When looking at the mass fractions of inorganic species during the cold season, I noticed that ammonium nitrate plays an important role. Correct me if I am wrong, but likely this hygroscopic PM constituent further promoted the water uptake

resulting in a correlation between aerosol liquid water and PM nitrate mass fraction. As higher nitrate mass fractions promote $CO_2^+$ release from the vaporizer, some $CO_2^+$ could be attributed to this artefact. Because the aqueous BBOA correlates with high aerosol liquid water content, some of the $CO_2^+$ detected during this time might be attributed to aqueous BBOA mass. This might lead to an overestimation of the aqueous BBOA concentration.

I suggest you quantify the importance of this artefact with your instrument. Importantly, if the artefact is significant, you should consider additional PMF runs with new fragmentation table setups.

References:

Pieber, Simone M., et al. "Inorganic salt interference on $CO_2^+$ in aerodyne AMS and ACSM organic aerosol composition studies." *Environmental Science & Technology* 50.19 (2016): 10494-10503.

Freney, Evelyn, et al. "The second ACTRIS inter-comparison (2016) for Aerosol Chemical Speciation Monitors (ACSM): Calibration protocols and instrument performance evaluations." *Aerosol Science and Technology* (2019): 1-13.

---

## Author Response (AR2)

**Replies to Referee #3**

The authors would like to thank Anonymous Referee #3 for his/her comments. Please find below our responses (in black) after the referee comments (in blue). The changes in the revised manuscript are written in *italic*.

*PMF-analysis*
First, as PMF plays a central role in the aqueous BBOA quantification, the description of the PMF analysis should not by any means be hidden in the supplementary material and it should be thoroughly step by step explained. The PMF analysis process arises few concerns regarding a-values and the amount of repetitions performed (number of iterations) that seems not to be reported at all in the manuscript. I suggest you explicitly write down in the manuscript how many repetitions were conducted to make the readers aware of the statistical robustness of the solution. Currently, the vague description of the PMF analysis arises doubts of the solidness of the result.
For example, the justification of different a-values used in the multi-linear engine (ME-2) analysis is not clear. The high a-value used for HOA (a = 0.5) is exceptional, and I personally did not see why it was necessary. Previous studies suggest an a-value of approx. 10% for HOA. In contrary to this, the BBOA was constrained with an a-value of 5%. This is allows a rather low degree of variability for the BBOA mass spectrum in the PMF analysis despite the fact that BBOA is known to vary a lot depending on the burning material (even up to 30%). While the a-values chosen in the current study can in some cases be fully justified, the motivation still needs to be explicitly written down.
It is absolutely crucial that you carefully motivate the selections of the constraints (a-values) and the solutions (number of factors) in the manuscript main text. Please spend time on this to make sure your readers (and I) can follow your process. You can even create a flow chart to describe it, but try to avoid just transferring the table jungle from the SI to the main text, try to summarize the logic you had in decision making.

**Authors reply**: We would like to thank the Referee #3 for his/her important comments. We agree that a full treatment of the PMF methodology is key in this study. We acknowledge that the first version of the manuscript has not provided the full details and that this needs to be improved. We only partly agree with the Referee about the necessity of including the full description in the main text, because this study is also intended to be suitable for a broad readership. In order to present the PMF methodology, we will therefore take the following actions: 1) include a new description of PMF in the main text which will be step-by-step but concise and based on graphical tools (a flow chart will be presented in a new Figure 2), and 2) a comprehensive description with extended text will be incorporated in the Supplementary Information (new section S2). We report below the specific changes and additions to main text and SI:

Main text, P5 L24 (revised version):
*The standardized source apportionment strategy introduced in Crippa et al. (2014) was applied to the twelve individual HR-TOF-AMS datasets (8 from BO and 4 from SPC). The PMF analysis followed the iterative, step-by-step protocol reported in Figure 2. A comprehensive description of the PMF protocol and of the criteria for identifying best solutions followed in each campaign, together with specific metrics for every single factor analysis (number of iterations, number of factors chosen, Q and residuals diagnostic plots, constrained factor profiles and a-values if applied, etc.), are reported in the supplemental section S2.*

[Figure]

**Figure 2.** *Schematic step-by-step procedure of adopted source apportionment approach.*

**Supplementary Section S2, P3:**

*The standardized source apportionment strategy introduced in Crippa et al. (2014) is systematically applied to the 12 available HR-TOF-AMS datasets (8 from BO and 4 from SPC) following the sequential steps reported below:*

*1. Unconstrained run (PMF): in a first step, a range of unconstrained runs was examined: solutions from two to eight factors are investigated (applying three pseudo-random starting point -seeds- each, for a total of 21 unconstrained runs) for all the datasets in order to choose the most appropriate number of interpretable factors, that resulted to be campaign-specific and ranged from 3 up to 6 (depending on the season, the site and the number of interpretable OOA factors). The most appropriate number of factors was chosen based on the residual analysis (inspecting and minimizing both the Q-value and the possible presence of structure in the residual diurnal trends) together with the correlation analysis of the factors with each other both in terms of mass-spectral and time-dependent similarities (Ulbrich et al., 2009). This means that the best number of factor is established when further increasing the number of factors does not improve the interpretation of the data, as the new factor time series and spectral profiles are highly correlated with those extracted from lower order solutions and cannot be explicitly associated to distinct sources or processes.*

*2. Constraining only HOA mass spectrum: after the most reasonable number of factors was identified, the HOA mass spectrum was constrained in a range of a-values (i.e., a=0, 0.05, 0.1, 0.3, 0.5) in order to check its attribution and any possible erroneous mixing between sources. Moreover various numbers of factors close to the optimal were tested: for example if the best number of factors identified was 5, we run solutions with 4, 5 and 6 factors. For every a-value, the model was*

*initiated from three different pseudo-random starting points (seeds), yielding 45 total runs for each reference spectral profile constrained. We tested also different reference HOA factor profiles from ambient deconvolved spectra of the high-resolution aerosol mass spectral database (URL: http://cires.colorado.edu/jimenezgroup/ HRAMSsd/", Ulbrich et al., 2009). In particular, for HOA we employed reference profiles from Mohr et al. (2012) (obtained at Barcelona urban background site) and from Crippa et al. (2013a) (from Paris).*

*Crippa et al. (2014) (and most of the subsequent literature) suggested low a-values (e.g., a=0.05– 0.1) for HOA profiles, given usual low variability of this source profile in most of the studies. Nevertheless applying these low a-values to our datasets resulted often in two split HOA factors with very similar profiles and time series or in additional HOA/BBOA-mixed factors. Moreover solutions with higher a-value associated to HOA (a=0.5) maximized the correlation with external tracers of traffic emissions (i.e., NOx, BC, EC) and minimized the residuals associated with rush hours in the diurnal trend of the residuals (see Table S3 and S4) and for this reason were chosen.*

*3. Looking for BBOA (if not identified before or mixed with HOA or COA): BBOA reference profiles were constrained when a not clear separation between BBOA and other primary factors (HOA and COA) were found. First of all the BBOA reference spectrum from Mohr et al. (2012) was constrained alternatively alone (in a range of a-values =0, 0.05, 0.1, 0.3, 0.5) and together with the HOA reference profile (always from Mohr et al., 2012). When simultaneous constraining of BBOA and HOA were applied, the a-values were independently varied for HOA and BBOA (a-value =0, 0.05, 0.1, 0.3, 0.5, giving 25 a-value combinations). For every a-value combination the model was initiated from three different pseudo-random starting points (seeds), yielding 75+15=90 total runs. Again, together with different a-values, various numbers of factors were tested close to the optimal, in order to study any possible improvements of the solution in term of both the analysis of the residuals and the correlation of the factors with each other and with external tracers of traffic (i.e., NOx, BC, EC) and biomass burning (Levoglucosan) emissions.*

*Actually in our analysis we found an improvement in constraining BBOA only in two cases out of 12: BO_spring 2014 and SPC_spring 2013 campaigns. In these two cases we needed a strong constrain (a-value of 0.05) to see a better separation between BBOA and COA (in the case of BO_spring 2014) and HOA (in the case of SPC_spring 2013). This low a-value is not common for constraining BBOA for which, given the degree of variability that the BBOA spectrum can have depending on the burning material and systems, higher values (a-value = 0.3–0.5) are usually suggested. Anyway, in our cases, applying the suggested values we didn't obtain any significant improvement in the separation between BBOA and HOA or COA factors. Using the selected a-value of 0.05 instead we found a better correlation with external tracers in both cases (see Table S4).*

*4. Looking for COA: even if not suspected from the initial unconstrained analysis (looking the possible presence of meal hour peaks in the diurnals and inspecting the f55-f57 relative abundance as suggested by Mohr et al. (2012)), in any case an attempt of looking for COA factor was done for each campaign.*

*COA reference profiles from Mohr et al. (2012) and Crippa et al. (2013a) were alternatively constrained alone (in a range of a-values =0, 0.05, 0.1, 0.3, 0.5). Only when the unconstrained or this first COA constraining resulted in a possible COA contribution, then COA reference profiles were constrained together with HOA and BBOA profiles (always from Mohr et al., 2012).*

*When simultaneous constraining of COA and HOA were applied, the a-values were independently varied for HOA and COA (a-value =0, 0.05, 0.1, 0.3, 0.5, giving 25 a-value combinations). Also the same a-values were applied constraining COA together with both HOA and BBOA profiles, varying each independently (giving 105 a-values combinations).*

*Despite these efforts, in our analysis only in 2 cases out of 12 there was the suspicion of a COA contribution and only in one case (BO_spring 2014) this contribution was considered real in the end (based on its spectral profile similarity with references and on the presence of meal hour*

*peaks). For this campaign actually the chosen solution was leaving COA profile unconstrained because constraining the COA profile (both from Mohr et al, 2012 and Crippa et al., 2013a reference profiles) leaded to split COA factors only with variable amount of m/z 44.*

*The COA factor identified in BO_spring 2014 campaign shows an early lunch-time peak in the diurnal trend (peaking around 11-12) and an higher than usual contribution of m/z 44, which leave some doubts in the correct quantification of this COA contribution. We considered the hypotheses of a misleading mixing-source between COA and HOA, COA and BBOA and also between COA and OOA: we tested all the possible combination of constraining (only HOA, HOA+BBOA, HOA+COA, HOA+BBOA+COA), a number of a-values (a-value =0, 0.05, 0.1, 0.3, 0.5) for each of this combination and also for different numbers of factors (from 4 to 7), which resulted in strong increases of the residuals with a clear diurnal pattern peaking between 11-12 (in the case of a reduced number of factors) or in split/mixed HOA, BBOA and COA profiles. Eventually we opted for the solution that minimizes the uncertainty in the identification of the other two primary components (HOA and BBOA) and maximizes their correlation with external tracers. This mainly because the focus of our study is on BB-related factors and because COA represents in any case just a minor factor found in only one campaign. We acknowledge this issue, but we leave the deeper investigation of the peculiarity of this COA factor to other possible future studies.*

*ITERATIVELY. Residual analysis: for each step the residual plots were consulted in order to evaluate whether the constrained profile(s) has (have) caused structures in the residuals. If so, the constrained profiles were tested with a higher a-value or rejected.*

*Oxidized organic aerosol components (OOAs) factors were never constrained because their mass spectra are characterized by a greater variability with respect to the POA factors, reflecting the multiplicity of atmospheric secondary formation and transformation processes contributing to SOA formation and composition (Canonaco et al. 2015).*

*When an unconstrained PMF solution was considered as the optimal one, PMF solutions for multiple values of FPEAK are explored to test the rotational ambiguity of the results. Chosen the best number of factors, variable FPEAKs values (from -0.6 to +0.6, with 0.2 steps) were applied and the resulting Q values, scaled residuals, and factor profiles and time series were examined to select the optimum solution.*

*Optimum solutions were selected if they satisfied the following set of criteria:*
*1. $fCO_2+$ <0.04 in HOA and COA factor profiles (HOA based on Aiken et al., 2009; Mohr et al., 2012; Crippa et al., 2013a, 2014 and COA based on Crippa et al., 2013a, 2013b; Mohr et al., 2012), with the exception of SPC_fall 2011 due to the peculiar meteorological conditions further described in section 3.2;*
*2. HOA correlates significantly with NOx, BC and EC;*
*3. HOA correlates better with NOx than COA; BBOA correlates significantly with levoglucosan;*
*4. The concentration ratios between the main POA factors (HOA and BBOA) and tracer compounds (used as source-specific ratios) are in a reasonable range compared with values in literature;*
*5. COA has a diurnal trend characterized by meal hours peaks (lunch and dinner time).*

*The interpretation of the retrieved source apportionment factors as organic aerosol sources is based on the comparison of their mass spectral profiles with reference ones (Table S5, S6 and S7), on the correlations with external data (see Table S8) and on the investigation of their diurnal trends (see Figure 3 of the main text). For the PMF-results already discussed in other papers (i.e., BO_2013winter and SPC_2011fall, and SPC_2012summer campaigns) we refer the reader to the corresponding publications (i.e, Gilardoni et al., 2014 & 2016 and Sullivan et al., 2016).*

*Regarding the other datasets, details of the best solution chosen for each campaign are reported in the following figures.*

**CO2+ release from the AMS vaporizer due to ammonium nitrate**

Another important concern of mine is the CO2+ release from the AMS vaporizer coinciding with high ammonium nitrate loadings. This effect was introduced recently by Pieber et al. (2016) and investigated thoroughly with AMS-type instrumentation by Freney et al. (2019). My wonder is whether you quantified this effect with your instrumentation. How much CO2+ did you detect during your ionization efficiency calibrations? Did you modify your fragmentation table(s) and adjust the CO2+ introduced by ammonium nitrate?

When looking at the mass fractions of inorganic species during the cold season, I noticed that ammonium nitrate plays an important role. Correct me if I am wrong, but likely this hygroscopic PM constituent further promoted the water uptake resulting in a correlation between aerosol liquid water and PM nitrate mass fraction. As higher nitrate mass fractions promote CO2+ release from the vaporizer, some CO2+ could be attributed to this artefact. Because the aqueous BBOA correlates with high aerosol liquid water content, some of the CO2+ detected during this time might be attributed to aqueous BBOA mass. This might lead to an overestimation of the aqueous BBOA concentration.

I suggest you quantify the importance of this artefact with your instrument. Importantly, if the artefact is significant, you should consider additional PMF runs with new fragmentation table setups.

**Authors reply**: The Authors want to thank the Referee # 3 for the interesting comment. We quantified the interference of ammonium nitrate on the $CO_2^+$ signal in our instruments using the data from the ionization efficiency calibrations (IE-Cal) conducted during the campaigns. We calculated the relationship "$b$" of $CO_2^+$ and $NO_3$ signals from the Hi-res (PIKA) analysis of each IE-Cal dataset, following the criteria suggested by Pieber et al. (2016): $b$ is the slope of the orthogonal distance linear fit of the $CO_2^+$ and $NO_3$ signals in nitrate equivalent mass (i.e., using a relative ionization efficiency =1). Results of our estimations are reported in the histogram in Figure AR1 compared to the reference values identified in Pieber et al. (2016). Despite the variability between different instruments (measuring at BO and SPC respectively) and even between campaigns using the same instrument (higher variability for the AMS used at BO), the graph clearly shows how the $b$ values for our instruments are far below the acceptable limit set to +3.4% (the median value in Pieber et al., 2016, considered acceptable even for periods of high inorganic mass fractions like our fall/winter campaigns) and actually close to the 10[th] percentile (P10=+0.4%) identified in the same study and considered a good performance. Given these low bias values, we believed not necessary to correct the data or to introduce the bias in the error estimation of our PMF analyses.

[Figure]

**Figure AR1.** Magnitude of the CO2+ related to the NO3 signal described as the slope *b* introduced by Pieber et al. (2016). The parameter *b* is the slope derived from the orthogonal distance linear fit of the CO2+ vs NO3 signal (expressed in nitrate equivalent mass, i.e., RIE=1) from the ionization efficiency calibrations (IE-Cal) carried out during the SUPERSITO campaigns. Blue bars represent the mean values ($n_{IE-Cal}$=2-3) of *b* corresponding to each campaign. Black and dashed grey lines are the *b* statistical parameters (P50=median, P25=25[th] percentile, P10=10[th] percentile, respectively) derived by Pieber et al. (2016) and used as reference to estimate the relevance of the NO3-interference in the determination of OA.

However, in order to evaluate the possible corresponding overestimation of our OOA factors especially in the campaigns when $NO_3$ concentrations are more relevant ($NO_3$/OA ratios close to 1), we calculated the average non-OA $CO_2^+$ concentrations for each campaign (based on the $NO_3$ concentrations and the specific slopes *b* of each campaign reported in Figure AR1). Considering the $CO_2^+$ fragments completely apportioned into the OOAs components, we attributed all the non-OA $CO_2^+$ mass to the total OOAs fraction. Table AR1 reports the average non-OA $CO_2^+$ mass calculated for each campaign and also the possible relative contribution of this interference to the total OOA mass. This relative contribution spans in the range 0.1-1.3%, pointing to a negligible influence of $NO_3$ in the quantification of OOA in our datasets, even during the periods strongly impacted by $NO_3$.

Moreover, considering the worst possible scenario in which PMF attributes all the $NO_3$-interference to a single OOA factor, we calculated the possible relative contribution of the non-OA $CO_2^+$ to every factor (taking them separately). It is important to highlight that even if this is the maximum possible contribution of the $NO_3$-interference in the determination of every single OOA factor, in any case it results to be negligible: we found values ranging 1.5-7.2% during the fall/winter campaigns, when Nitrate concentrations in atmosphere are relevant ($NO_3$/OA ratios close to 1), and even lower during spring/summer campaigns. Even more important, the highest values are not related to the aqSOA factors (highlighted in yellow in the Table AR1) or to the OOA factors correlating with $NO_3$. So we can reasonably consider our estimation of the aqSOA factors (and also of the other OOA factors) not biased by the $NO_3$-interference.

**Table AR1.** Average non-OA $CO_2^+$ mass ($\mu g/m^3$) calculated for each campaign (based on the corresponding specific *b* value and Nitrate concentrations in atmosphere) and possible relative contribution of this artifact to the OOA mass fractions. Last column reports the ratio between average Nitrate and OA concentrations (NO3/OA) measured in each campaign as an indication of the variable importance of the inorganic mass fraction in the different campaigns.

| | non-OA CO2+ (µg/m3) | Relative contr. to | | | | | |
| | | OOA_TOT | OOA1 | OOA2 | OOA3 | OOA4 | NO3/OA |
|---|---|---|---|---|---|---|---|
| **2011_BO_fall. (nov.-dic.)** | 0.07 | 1.1% | 1.9% | 2.4% | - | - | 0.77 |
| **2012_BO_summer (jun-jul.)** | 0.00 | 0.1% | 0.1% | 0.1% | - | - | 0.10 |
| **2012_BO_fall (oct.-nov.)** | 0.02 | 0.6% | 3.2% | 1.6% | 1.5% | - | 0.74 |
| **2013_BO_winter (jan.-feb.)** | 0.06 | 1.1% | 2.9% | 3.4% | 3.7% | - | 0.83 |
| **2013_BO_spring (may)** | 0.02 | 1.2% | 3.8% | 6.1% | 2.4% | - | 0.61 |
| **2013_BO_fall (oct.)** | 0.03 | 1.3% | 5.5% | 4.0% | 2.8% | - | 1.17 |
| **2014_BO_winter (jan.-feb.)** | 0.02 | 1.0% | 7.2% | 1.8% | 3.2% | - | 1.07 |
| **2014_BO_spring (may)** | 0.01 | 0.3% | 0.9% | 1.0% | 0.7% | - | 0.20 |
| **2011_SPC_fall (nov.-dic.)** | 0.04 | 1.1% | 1.1% | - | - | - | 0.66 |
| **2012_SPC_summer (jun.-jul.)** | 0.00 | 0.1% | 0.5% | 0.2% | 0.2% | 0.1% | 0.26 |
| **2013_SPC_spring (may)** | 0.01 | 0.4% | 1.2% | 2.6% | 0.8% | - | 0.96 |
| **2013_SPC_fall (oct.)** | 0.01 | 0.4% | 1.2% | 2.1% | 1.0% | - | 0.80 |

Authors add a short description of this evaluation to the main text in order to make the reader aware of the tests performed, but without additional figures or tables.

**Main text, P4, L33** (revised version):
*Data from IE calibrations were also used to quantify the interference of ammonium nitrate on the CO2+ signal for the different instruments and campaigns following the criteria suggested by Pieber et al. (2016). The relationship "b" (the slope of the orthogonal distance linear fit of the $CO_2^+$ and $NO_3^-$ signals expressed in nitrate equivalent mass, i.e., RIE=1) in our estimations resulted spanning between +0.2% and +1.4% (+0.65±0.35% on average), well below the limit considered acceptable even for periods of high inorganic mass fractions set to +3.4% (Pieber et al., 2016).*

**List of relevant changes**

Relevant changes made in the revised Manuscript (as already discussed in the Replies to the Referee #3 and tracked in the following marked-up version) are listed below.

**Main text, P5 L24 & Supplementary Section S2, P3** (revised version): Following the suggestion of Referee #3, the Authors improved the treatment of the PMF methodology both in the main text (including a new description of PMF approach step-by-step but concise and based on a flow chart reported in a new Figure 2), and in the Supplementary Information (adding a comprehensive description with extended text).

**Main text, P4, L33** (revised version): Following the suggestion of Referee #3, the Authors add a short description of the evaluation of the interference of NO3 on the CO2+ signal to the main text in order to make the reader aware of the tests performed, but without additional figures or tables.

[revised manuscript text omitted]

Marco Paglione 15/11/y 10:01